# AdjointDEIS: Efficient Gradients for Diffusion Models

**Zander W. Blasingame**
Clarkson University
blasinzw@clarkson.edu

**Chen Liu**
Clarkson University
cliu@clarkson.edu

## Abstract

The optimization of the latents and parameters of diffusion models with respect to some differentiable metric defined on the output of the model is a challenging and complex problem. The sampling for diffusion models is done by solving either the *probability flow* ODE or diffusion SDE wherein a neural network approximates the score function allowing a numerical ODE/SDE solver to be used. However, naïve backpropagation techniques are memory intensive, requiring the storage of all intermediate states, and face additional complexity in handling the injected noise from the diffusion term of the diffusion SDE. We propose a novel family of bespoke ODE solvers to the continuous adjoint equations for diffusion models, which we call *AdjointDEIS*. We exploit the unique construction of diffusion SDEs to further simplify the formulation of the continuous adjoint equations using *exponential integrators*. Moreover, we provide convergence order guarantees for our bespoke solvers. Significantly, we show that the continuous adjoint equations for diffusion SDEs actually simplify to a simple ODE. Lastly, we demonstrate the effectiveness of AdjointDEIS for guided generation with an adversarial attack in the form of the face morphing problem. Our code will be released at https://github.com/zblasingame/AdjointDEIS.

## 1 Introduction

Diffusion models are a large family of state-of-the-art generative models which learn to map samples drawn from Gaussian white noise into the data distribution [1, 2]. These diffusion models have achieved state-of-the-art performance on prominent tasks such as image generation [3–5], audio generation [6, 7], or video generation [8]. Often, state-of-the-art models are quite large and training them is prohibitively expensive [9]. As such, it is fairly common to adapt a pre-trained model to a specific task for post-training. In this way, the generative model can learn new concepts, identities, or tasks without having to train the entire model [10–12]. Additional work has also proposed algorithms for guiding the generative process of diffusion models [13, 14].

One method of guiding or directing the generative process is to solve an optimization problem w.r.t. some guidance function $\mathcal{L}$ defined in the image space $\mathbb{R}^d$. This guidance function works on the output of the diffusion model and assesses how "good" the output is. However, the diffusion model works by iteratively removing noise until a clean sample is reached. As such, we need to be able to efficiently backpropagate gradients through the entire generative process. As Song et al. [15] showed, the diffusion SDE can be simplified to an associated ODE, and as such, many efficient ODE/SDE solvers have been developed for diffusion models [16–18]. However, naïvely applying backpropagation to the diffusion model is inflexible and memory intensive; moreover, such an approach is not trivial to apply to the diffusion models that used an SDE solver instead of an ODE solver.

38th Conference on Neural Information Processing Systems (NeurIPS 2024).

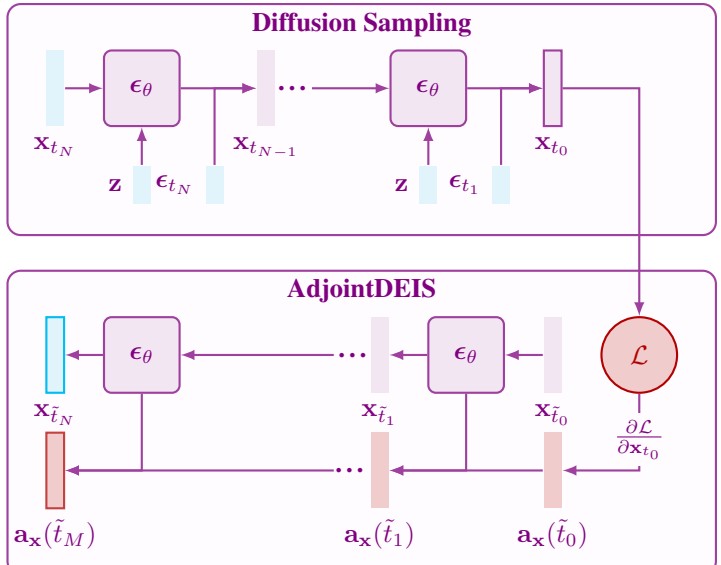

Figure 1: A high-level overview of the AdjointDEIS solver to the continuous adjoint equations for diffusion models. The sampling schedule consists of $\{t_n\}_{n=0}^N$ timesteps for the diffusion model and $\{\tilde{t}_n\}_{n=0}^M$ timesteps for AdjointDEIS. The gradients $\mathbf{a_x}(T)$ can be used to optimize $\mathbf{x}_T$ to find some optimal $\mathbf{x}_T^*$.

## 1.1 Contributions

Inspired by the work of Chen et al. [19] we study the application of continuous adjoint equations to diffusion models, with a focus on training-free guided generation with diffusion models. We introduce several theoretical contributions and technical insights to both improve the ability to perform certain guided generation tasks and to gain insight into guided generation with diffusion models.

First, we introduce *AdjointDEIS* a bespoke family of ODE solvers which can efficiently solve the continuous adjoint equations for both diffusion ODEs and SDEs. Moreover, we show that the continuous adjoint equations for diffusion SDEs simplify to a mere ODE. Next, we show how to calculate the continuous adjoint equation for conditional information which evolves with *time* (rather than being constant). To the best of our knowledge, we are the first to consider conditional information which evolves with time for neural ODEs. Overall, multiple theoretical contributions and technical insights are provided to bring a new family of techniques for the guided generation of diffusion models, which we evaluate experimentally on the task of face morphing.

## 1.2 Diffusion Models

In this subsection, we provide a brief overview of diffusion models. Diffusion models learn a generative process by first perturbing the data distribution into an isotropic Gaussian by progressively adding Gaussian noise to the data distribution. Then a neural network is trained to perform denoising steps, allowing for sampling of the data distribution via sampling of a Gaussian distribution [2, 9]. Assume that we have an $d$-dimensional random variable $\mathbf{x} \in \mathbb{R}^d$ with some distribution $p_{\text{data}}(\mathbf{x})$. Then diffusion models begin by diffusing $p_{\text{data}}(\mathbf{x})$ according to the diffusion SDE [2], an Itô SDE given as

$$\mathrm{d}\mathbf{x}_t = f(t)\mathbf{x}_t \, \mathrm{d}t + g(t) \, \mathrm{d}\mathbf{w}_t \tag{1.1}$$

where $t \in [0, T]$ denotes time with fixed constant $T > 0$, $f(\cdot)$ and $g(\cdot)$ denote the drift and diffusion coefficients, and $\mathbf{w}_t$ denotes the standard Wiener process. The trajectories of $\mathbf{x}_t$ follow the distributions $p_t(\mathbf{x}_t)$ with $p_0(\mathbf{x}_0) \equiv p_{\text{data}}(\mathbf{x})$ and $p_T(\mathbf{x}_T) \approx \mathcal{N}(\mathbf{0}, \mathbf{I})$. Under some regularity conditions Song et al. [15] showed that Equation (1.1) has a reverse process as time runs backwards from $T$ to $0$ with initial marginal distribution $p_T(\mathbf{x}_T)$ governed by

$$\mathrm{d}\mathbf{x}_t = [f(t)\mathbf{x}_t - g^2(t)\nabla_\mathbf{x} \log p_t(\mathbf{x}_t)] \, \mathrm{d}t + g(t) \, \mathrm{d}\bar{\mathbf{w}}_t \tag{1.2}$$

where $\bar{\mathbf{w}}_t$ is the standard Wiener process as time runs backwards. Solving Equation (1.2) is what allows diffusion models to draw samples from $p_{\text{data}}(\mathbf{x})$ by sampling $p_T(\mathbf{x}_T)$. The unknown term in Equation (1.2) is the *score function* $\nabla_{\mathbf{x}} \log p_t(\mathbf{x}_t)$, which in practice is modeled by a neural network that estimates the scaled score function, $\boldsymbol{\epsilon}_\theta(\mathbf{x}_t, t) \approx -\sigma_t \nabla_{\mathbf{x}} \log p_t(\mathbf{x}_t)$, or some closely related quantity like $\mathbf{x}_0$-prediction [1, 2, 20].

**Probability Flow ODE.** The practical choice of a step size when discretizing SDEs is limited by the randomness of the Wiener process as a large step size, *i.e.*, a small number of steps, can cause non-convergence, particularly in high-dimensional spaces [16]. Sampling an equivalent Ordinary Differential Equation (ODE) over an SDE would enable faster sampling. Song et al. [15] showed there exists an Ordinary Differential Equation (ODE) whose marginal distribution at time $t$ is identical to that of Equation (1.2) given as

$$\frac{\mathrm{d}\mathbf{x}_t}{\mathrm{d}t} = f(t)\mathbf{x}_t - \frac{1}{2}g^2(t)\nabla_{\mathbf{x}} \log p_t(\mathbf{x}_t). \tag{1.3}$$

The ODE in Equation (1.3) is known as the *probability flow ODE* [15]. As the noise prediction network, $\boldsymbol{\epsilon}_\theta(\mathbf{x}_t, t)$, is trained to model the scaled score function, Equation (1.3) can be parameterized as

$$\frac{\mathrm{d}\mathbf{x}_t}{\mathrm{d}t} = f(t)\mathbf{x}_t + \frac{g^2(t)}{2\sigma_t}\boldsymbol{\epsilon}_\theta(\mathbf{x}_t, t) \tag{1.4}$$

w.r.t. the noise prediction network. For brevity, we refer to this as a diffusion ODE.

Although there exist several popular choices for the drift and diffusion coefficients, we opt to use the *de facto* choice which is known as the Variance Preserving (VP) type diffusion SDE [1, 15, 21]. The coefficients for VP-type SDEs are given as

$$f(t) = \frac{\mathrm{d}\log\alpha_t}{\mathrm{d}t}, \quad g^2(t) = \frac{\mathrm{d}\sigma_t^2}{\mathrm{d}t} - 2\frac{\mathrm{d}\log\alpha_t}{\mathrm{d}t}\sigma_t^2, \tag{1.5}$$

which corresponds to sampling $\mathbf{x}_t$ from the distribution $q(\mathbf{x}_t \mid \mathbf{x}_0) = \mathcal{N}(\alpha_t\mathbf{x}_0, \sigma_t^2\mathbf{I})$.

## 2 Adjoint Diffusion ODEs

**Problem statement.** Given the diffusion ODE in Equation (1.4), we wish to solve the following optimization problem:

$$\underset{\mathbf{x}_T, \mathbf{z}, \theta}{\arg\min} \ \mathcal{L}\left(\mathbf{x}_T + \int_T^0 f(t)\mathbf{x}_t + \frac{g^2(t)}{2\sigma_t}\boldsymbol{\epsilon}_\theta(\mathbf{x}_t, \mathbf{z}, t) \ \mathrm{d}t\right). \tag{2.1}$$

*I.e.*, we seek to find the optimal $\mathbf{x}_T$, $\mathbf{z}$, and $\theta$ that satisfies our guidance function $\mathcal{L}$. *N.B.*, the noise-prediction model is conditioned on additional information $\mathbf{z}$.

Unlike GANs which can update the latent representation through GAN inversion [22, 23], as seen in Equation (2.1) diffusion models require more care as they model an ODE or SDE and require numerical solvers. Therefore, to update the latent representation, model parameters, and conditional information, we must backpropagate the gradient of loss defined on the output, $\partial\mathcal{L}(\mathbf{x}_0)/\partial\mathbf{x}_0$ through the entire ODE or SDE.

A key insight of this work is the connection between the adjoint ODE used in neural ODEs by Chen et al. [19] and specialized ODE/SDE solvers by [16–18] for diffusion models. It has been well observed that diffusion models are a type of neural ODE [15, 24]. Since a diffusion model can be thought of as a neural ODE, we can solve the continuous adjoint equations [25] to find useful gradients for guided generation. We can then exploit the unique structure of diffusion models to develop efficient bespoke ODE solvers for the continuous adjoint equations.

Let $\boldsymbol{f}_\theta$ describe a parameterized neural field of the probability flow ODE, *i.e.*, the R.H.S of Equation (1.4), defined as

$$\boldsymbol{f}_\theta(\mathbf{x}_t, \mathbf{z}, t) = f(t)\mathbf{x}_t + \frac{g^2(t)}{2\sigma_t}\boldsymbol{\epsilon}_\theta(\mathbf{x}_t, \mathbf{z}, t). \tag{2.2}$$

Then $\boldsymbol{f}_\theta(\mathbf{x}_t, \mathbf{z}, t)$ describes a neural ODE which admits an adjoint state, $\mathbf{a}_{\mathbf{x}} := \partial\mathcal{L}/\partial\mathbf{x}_t$ (and likewise for $\mathbf{a}_{\mathbf{z}}(t)$ and $\mathbf{a}_\theta(t)$), which solve the continuous adjoint equations [25, Theorem 5.2] in the form of

the following Initial Value Problem (IVP):

$$\mathbf{a_x}(0) = \frac{\partial \mathcal{L}}{\partial \mathbf{x}_0}, \qquad\qquad \frac{\mathrm{d}\mathbf{a_x}}{\mathrm{d}t}(t) = -\mathbf{a_x}(t)^\top \frac{\partial \boldsymbol{f}_\theta(\mathbf{x}_t, \mathbf{z}, t)}{\partial \mathbf{x}_t},$$

$$\mathbf{a_z}(0) = \mathbf{0}, \qquad\qquad \frac{\mathrm{d}\mathbf{a_z}}{\mathrm{d}t}(t) = -\mathbf{a_x}(t)^\top \frac{\partial \boldsymbol{f}_\theta(\mathbf{x}_t, \mathbf{z}, t)}{\partial \mathbf{z}},$$

$$\mathbf{a_\theta}(0) = \mathbf{0}, \qquad\qquad \frac{\mathrm{d}\mathbf{a_\theta}}{\mathrm{d}t}(t) = -\mathbf{a_x}(t)^\top \frac{\partial \boldsymbol{f}_\theta(\mathbf{x}_t, \mathbf{z}, t)}{\partial \theta}. \qquad (2.3)$$

We refer to this system of equations in Equation (2.3) as the *adjoint diffusion ODE*[1] as it describes the continuous adjoint equations for the empirical probability flow ODE.

*N.B.*, in the literature of diffusion models, the sampling process is often done in reverse time *i.e.*, the initial noise is $\mathbf{x}_T$ and the final sample is $\mathbf{x}_0$. Due to this convention, solving the adjoint diffusion ODE *backwards* actually means integrating *forwards* in time. Thus, while diffusion models learn to compute $\mathbf{x}_t$ from $\mathbf{x}_s$ with $s > t$, the adjoint diffusion ODE seeks to compute $\mathbf{a_x}(s)$ from $\mathbf{a_x}(t)$.

## 2.1 Simplified Formulation of the Empirical Adjoint Probability Flow ODE

We show that rather than treating $\boldsymbol{f}_\theta$ as a black box, the specific structure of the probability flow ODE is carried over to the adjoint probability flow ODE, allowing the adjoint probability flow ODE to be simplified into a special exact formulation.

By evaluating the gradient of $\boldsymbol{f}_\theta$ w.r.t. $\mathbf{x}_t$ for each term in Equation (2.2) we can rewrite the adjoint diffusion ODE for $\mathbf{a_x}(t)$ in Equation (2.3) as

$$\frac{\mathrm{d}\mathbf{a_x}}{\mathrm{d}t}(t) = -f(t)\mathbf{a_x}(t) - \frac{g^2(t)}{2\sigma_t}\mathbf{a_x}(t)^\top \frac{\partial \boldsymbol{\epsilon}_\theta(\mathbf{x}_t, \mathbf{z}, t)}{\partial \mathbf{x}_t}. \qquad (2.4)$$

Due to the gradient of the drift term in Equation (2.4), further manipulations are required to put the empirical adjoint probability flow ODE into a sufficiently "nice" form. We follow the approach used by [16, 18] to simplify the empirical probability flow ODE with the use of exponential integrators and a change of variables. By applying the integrating factor $\exp\left(\int_0^t f(\tau)\,\mathrm{d}\tau\right)$ to Equation (2.4), we find:

$$\frac{\mathrm{d}}{\mathrm{d}t}\left[e^{\int_0^t f(\tau)\,\mathrm{d}\tau}\mathbf{a_x}(t)\right] = -e^{\int_0^t f(\tau)\,\mathrm{d}\tau}\frac{g^2(t)}{2\sigma_t}\mathbf{a_x}(t)^\top \frac{\partial \boldsymbol{\epsilon}_\theta(\mathbf{x}_t, \mathbf{z}, t)}{\partial \mathbf{x}_t}. \qquad (2.5)$$

Then, the exact solution at time $s$ given time $t < s$ is found to be

$$\mathbf{a_x}(s) = \underbrace{e^{\int_s^t f(\tau)\,\mathrm{d}\tau}\mathbf{a_x}(t)}_{\text{linear}} - \underbrace{\int_t^s e^{\int_s^u f(\tau)\,\mathrm{d}\tau}\frac{g^2(u)}{2\sigma_u}\mathbf{a_x}(u)^\top \frac{\boldsymbol{\epsilon}_\theta(\mathbf{x}_u, \mathbf{z}, u)}{\partial \mathbf{x}_u}\,\mathrm{d}u}_{\text{non-linear}}. \qquad (2.6)$$

Like with solvers for diffusion models which leverage exponential integrators, we are able to transform the adjoint diffusion ODE into a non-stiff form by separating the linear and non-linear component. Moreover, we can compute the linear in closed form, thereby *eliminating* the discretization error in the linear term. However, we still need to approximate the non-linear term which consists of a difficult integral about the complex noise-prediction model. This is where the insight of Lu et al. [16] to integrate in the log-SNR domain becomes invaluable. Let $\lambda_t := \log(\alpha_t/\sigma_t)$ be one half of the log-SNR. Then, with using this new variable and computing the drift and diffusion coefficients in closed form, we can rewrite Equation (2.6) as

$$\mathbf{a_x}(s) = \frac{\alpha_t}{\alpha_s}\mathbf{a_x}(t) + \frac{1}{\alpha_s}\int_t^s \alpha_u \sigma_u \frac{\mathrm{d}\lambda_u}{\mathrm{d}u}\mathbf{a_x}(u)^\top \frac{\boldsymbol{\epsilon}_\theta(\mathbf{x}_u, \mathbf{z}, u)}{\partial \mathbf{x}_u}\,\mathrm{d}u. \qquad (2.7)$$

As $\lambda_t$ is a strictly decreasing function w.r.t. $t$ it therefore has an inverse function $t_\lambda$ that satisfies $t_\lambda(\lambda_t) = t$, and, with abuse of notation, we let $\mathbf{x}_\lambda := \mathbf{x}_{t_\lambda(\lambda)}$, $\mathbf{a_x}(\lambda) := \mathbf{a_x}(t_\lambda(\lambda))$, &c. and let the reader infer from context if the function is mapping the log-SNR back into the time domain or already in the time domain. Then by rewriting Equation (2.7) as an exponentially weighted integral and performing an analogous derivation for $\mathbf{a_z}(t)$ and $\mathbf{a_\theta}(t)$, we arrive at the following.

---

[1]A more precise and technical name would be the *empirical adjoint probability flow ODE*; however, for brevity's sake we prefer the adjoint diffusion ODE and leave the reader to infer from context that this describes an empirical model with learned score function.

**Proposition 2.1** (Exact solution of adjoint diffusion ODEs). *Given initial values* $[\mathbf{a_x}(t), \mathbf{a_z}(t), \mathbf{a}_\theta(t)]$ *at time* $t \in (0, T)$*, the solution* $[\mathbf{a_x}(s), \mathbf{a_z}(s), \mathbf{a}_\theta(s)]$ *at time* $s \in (t, T]$ *of adjoint diffusion ODEs in Equation* (2.3) *is*

$$\mathbf{a_x}(s) = \frac{\alpha_t}{\alpha_s}\mathbf{a_x}(t) + \frac{1}{\alpha_s}\int_{\lambda_t}^{\lambda_s} \alpha_\lambda^2 e^{-\lambda}\mathbf{a_x}(\lambda)^\top \frac{\partial \boldsymbol{\epsilon}_\theta(\mathbf{x}_\lambda, \mathbf{z}, \lambda)}{\partial \mathbf{x}_\lambda} \, \mathrm{d}\lambda, \tag{2.8}$$

$$\mathbf{a_z}(s) = \mathbf{a_z}(t) + \int_{\lambda_t}^{\lambda_s} \alpha_\lambda e^{-\lambda}\mathbf{a_x}(\lambda)^\top \frac{\partial \boldsymbol{\epsilon}_\theta(\mathbf{x}_\lambda, \mathbf{z}, \lambda)}{\partial \mathbf{z}} \, \mathrm{d}\lambda, \tag{2.9}$$

$$\mathbf{a}_\theta(s) = \mathbf{a}_\theta(t) + \int_{\lambda_t}^{\lambda_s} \alpha_\lambda e^{-\lambda}\mathbf{a_x}(\lambda)^\top \frac{\partial \boldsymbol{\epsilon}_\theta(\mathbf{x}_\lambda, \mathbf{z}, \lambda)}{\partial \theta} \, \mathrm{d}\lambda. \tag{2.10}$$

The complete derivations of Proposition 2.1 can be found in Appendix B.1.

There is a nice symmetry between Equations (2.8) to (2.10), while the adjoint of the solution trajectories evolves with a weighting of $\alpha_t/\alpha_s$ in the linear term and the integral term is weighted by $\alpha_t^2/\alpha_s^2$ reflecting the double partial $\partial\mathbf{x}_t$ in the adjoint and Jacobian terms. Conversely, the adjoint state for the conditional information and model parameters evolves with no weighting on the linear term and the integral is only weighted by $\alpha_t/\alpha_s$. This follows from the vector fields being independent of $\mathbf{a_z}$ and $\mathbf{a}_\theta$. These equations, while reflecting the special nature of this formulation of diffusion models, also have an appealing parallel with the exact solution for diffusion ODEs Lu et al. [16, Proposition 3.1].

## 2.2 Numerical Solvers for AdjointDEIS

The numerical solver for the adjoint empirical probability flow ODE, now in light of Equation (2.8), only needs to focus on approximating the exponentially weighted integral of $\boldsymbol{\epsilon}_\theta$ from $\lambda_t$ to $\lambda_s$, a well-studied problem in the literature on exponential integrators [26, 27]. To approximate this integral, we evaluate the Taylor expansion of the Jacobian vector product to further simplify the ODE. For notational convenience let $\mathbf{V}(\mathbf{x};t)$ denote the scaled vector-Jacobian product of the adjoint state $\mathbf{a_x}(t)$ and the gradient of the model w.r.t. $\mathbf{x}_t$, *i.e.*,

$$\mathbf{V}(\mathbf{x};t) = \alpha_t^2\mathbf{a_x}(t)^\top\frac{\partial\boldsymbol{\epsilon}_\theta(\mathbf{x}_t, \mathbf{z}, t)}{\partial\mathbf{x}_t}, \tag{2.11}$$

and likewise we let $\mathbf{V}^{(n)}(\mathbf{x};\lambda)$ denote the $n$-th derivative w.r.t. to $\lambda$. For $k \geq 1$, the $(k-1)$-th Taylor expansion at $\lambda_t$ is

$$\mathbf{V}(\mathbf{x};\lambda) = \sum_{n=0}^{k-1}\frac{(\lambda-\lambda_t)^n}{n!}\mathbf{V}^{(n)}(\mathbf{x};\lambda_t) + \mathcal{O}((\lambda-\lambda_t)^k). \tag{2.12}$$

Plugging this expansion into Equation (2.8) and letting $h = \lambda_s - \lambda_t$ yields

$$\mathbf{a_x}(s) = \underbrace{\frac{\alpha_t}{\alpha_s}\mathbf{a_x}(t)}_{\substack{\text{Linear term}\\ \textbf{Exactly computed}}} + \frac{1}{\alpha_s}\sum_{n=0}^{k-1}\underbrace{\mathbf{V}^{(n)}(\mathbf{x};\lambda_t)}_{\substack{\text{Derivatives}\\ \textbf{Approximated}}}\underbrace{\int_{\lambda_t}^{\lambda_s}\frac{(\lambda-\lambda_t)^n}{n!}e^{-\lambda}\,\mathrm{d}\lambda}_{\substack{\text{Coefficients}\\ \textbf{Analytically computed}}} + \underbrace{\mathcal{O}(h^{k+1})}_{\substack{\text{Higher-order errors}\\ \textbf{Omitted}}}. \tag{2.13}$$

With this expansion, the number of terms which need to be estimated is further reduced as the exponentially weighted integral $\int_{\lambda_t}^{\lambda_s}\frac{(\lambda-\lambda_t)^n}{n!}e^{-\lambda}\,\mathrm{d}\lambda$ can be solved **analytically** by applying $n$ times integration-by-parts [16, 28]. Therefore, the only errors in solving this ODE occur in the approximation of the $n$-th order total derivatives of the vector-Jacobian product and the higher-order error terms $\mathcal{O}(h^{k+1})$. By dropping the $\mathcal{O}(h^{k+1})$ error term and approximating the first $(k-1)$-th derivatives of the vector-Jacobian product, we can derive $k$-th order solvers for adjoint diffusion ODEs. We decide to name such solvers as *Adjoint Diffusion Exponential Integrator Sampler (AdjointDEIS)* reflecting our use of the *exponential integrator* to simplify the ODEs and pay homage to DEIS from [18] that explored the use of exponential integrators for diffusion ODEs. Consider the case of $k = 1$, by dropping the error term $\mathcal{O}(h^2)$ we construct the AdjointDEIS-1 solver with the following algorithm.

**AdjointDEIS-1.** Given an initial augmented adjoint state $[\mathbf{a_x}(t), \mathbf{a_z}(t), \mathbf{a}_\theta(t)]$ at time $t \in (0, T)$, the solution $[\mathbf{a_x}(s), \mathbf{a_z}(s), \mathbf{a}_\theta(s)]$ at time $s \in (t, T]$ is approximated by

$$
\mathbf{a_x}(s) = \frac{\alpha_t}{\alpha_s}\mathbf{a_x}(t) + \sigma_s(e^h - 1)\frac{\alpha_t^2}{\alpha_s^2}\mathbf{a_x}(t)^\top \frac{\partial \boldsymbol{\epsilon}_\theta(\mathbf{x}_t, \mathbf{z}, t)}{\partial \mathbf{x}_t},
$$

$$
\mathbf{a_z}(s) = \mathbf{a_z}(t) + \sigma_s(e^h - 1)\frac{\alpha_t}{\alpha_s}\mathbf{a_x}(t)^\top \frac{\partial \boldsymbol{\epsilon}_\theta(\mathbf{x}_t, \mathbf{z}, t)}{\partial \mathbf{z}},
$$

$$
\mathbf{a}_\theta(s) = \mathbf{a}_\theta(t) + \sigma_s(e^h - 1)\frac{\alpha_t}{\alpha_s}\mathbf{a_x}(t)^\top \frac{\partial \boldsymbol{\epsilon}_\theta(\mathbf{x}_t, \mathbf{z}, t)}{\partial \theta}. \tag{2.14}
$$

Higher-order expansions of Equation (2.12) require estimations of the $n$-th order derivatives of the vector Jacobian product which can be approximated via *multi-step* methods, such as Adams-Bashforth methods [29]. This has the added benefit of reduced computational overhead, as the multi-step method just reuses previous values to approximate the higher-order derivatives. Moreover, multi-step methods are empirically more efficient than single-step methods [29]. Combining the Taylor expansions in Equation (2.12) with techniques for designing multi-step solvers, we propose a novel multi-step second-order solver for the adjoint empirical probability flow ODE which we call *AdjointDEIS-2M*. This algorithm combines the previous values of the vector Jacobian product at time $t$ and time $r$ to predict $\mathbf{a}_s$ *without* any additional intermediate values.

**AdjointDEIS-2M.** We assume having a previous solution $\mathbf{a_x}(r)$ and model output $\boldsymbol{\epsilon}_\theta(\mathbf{x}_r, \mathbf{z}, r)$ at time $r < t < s$, let $\rho$ denote $\rho = \frac{\lambda_t - \lambda_r}{h}$. Then the solution $\mathbf{a}_s$ at time $s$ to Equation (2.4) is estimated to be

$$
\mathbf{a_x}(s) = \frac{\alpha_t}{\alpha_s}\mathbf{a_x}(t) + \sigma_s(e^h - 1)\frac{\alpha_t^2}{\alpha_s^2}\mathbf{a_x}(t)^\top \frac{\partial \boldsymbol{\epsilon}_\theta(\mathbf{x}_t, \mathbf{z}, t)}{\partial \mathbf{x}_t}
$$

$$
+ \sigma_s \frac{e^h - 1}{2\rho}\left(\frac{\alpha_t^2}{\alpha_s^2}\mathbf{a_x}(t)^\top \frac{\partial \boldsymbol{\epsilon}_\theta(\mathbf{x}_t, \mathbf{z}, t)}{\partial \mathbf{x}_t} - \frac{\alpha_r^2}{\alpha_s^2}\mathbf{a_x}(r)^\top \frac{\partial \boldsymbol{\epsilon}_\theta(\mathbf{x}_r, \mathbf{z}, r)}{\partial \mathbf{x}_r}\right). \tag{2.15}
$$

For brevity, we omit the details of the AdjointDEIS-2M solver for $\mathbf{a_z}(t)$ and $\mathbf{a}_\theta(t)$; rather, we provide the complete derivation and details in Appendix B. Likewise, the full algorithm can be found in Appendix G.1. The advantage of a higher-order solver is that it is generally more efficient, requiring fewer steps due to its higher convergence order. We show that AdjointDEIS-$k$ is a $k$-th order solver, as stated in the following theorem. The proof is in Appendix C.

**Theorem 2.1** (AdjointDEIS-$k$ as a $k$-th order solver). *Assume the function $\boldsymbol{\epsilon}_\theta(\mathbf{x}_t, \mathbf{z}, t)$ and its associated vector-Jacobian products follow the regularity conditions detailed in Appendix C, then for $k = 1, 2$, AdjointDEIS-$k$ is a $k$-th order solver for adjoint diffusion ODEs, i.e., for the sequence $\{\tilde{\mathbf{a}}_\mathbf{x}(t_i)\}_{i=1}^M$ computed by AdjointDEIS-$k$, the global truncation error at time $T$ satisfies $\tilde{\mathbf{a}}_\mathbf{x}(t_M) - \mathbf{a_x}(T) = \mathcal{O}(h_{max}^2)$, where $h_{max} = \max_{1 \leq j \leq M}(\lambda_{t_i} - \lambda_{t_{i-1}})$. Likewise, AdjointDEIS-$k$ is a $k$-th order solver for the estimated gradients w.r.t. $\mathbf{z}$ and $\theta$.*

As previous work has shown that higher-order solvers may be unsuitable for large guidance scales [16–18] we do explicitly construct or analyze any solvers for $k > 2$ and leave such explorations for future study.

## 2.3 Scheduled Conditional Information

Thus far, we have held the conditional information constant across time, *i.e.*, at each time $t \in [0, T]$ the conditional information supplied to the neural network is $\mathbf{z}$. What if, however, we had some scheduled conditional information $\mathbf{z}_t$? We show that with some mild assumptions, using scheduled conditional information $\mathbf{z}_t$ does not actually change the continuous adjoint equation for $\mathbf{z}_t$ from the equations derived from $\mathbf{z}$ sans a substitution of $\mathbf{z}_t$ with $\mathbf{z}$.

While motivated by the case of scheduled conditional information in guided generation with diffusion models, this result applies to neural ODEs more generally, which could open future research directions. We state this result more formally in Theorem 2.2 with the proof in Appendix D. Note, as this applies more generally than to just AdjointDEIS, so we express this result for some arbitrary neural ODE with vector field $\boldsymbol{f}_\theta(\mathbf{x}_t, \mathbf{z}_t, t)$ and use the forward-time flow convention rather than the reverse-time convention of diffusion models.

**Theorem 2.2.** *Suppose there exists a function* $\mathbf{z} : [0, T] \to \mathbb{R}^z$ *which can be defined as a càdlàg*[2] *piecewise function where* $\mathbf{z}$ *is continuous on each partition of* $[0, T]$ *given by* $\Pi = \{0 = t_0 < t_1 < \cdots < t_n = T\}$ *and whose right derivatives exist for all* $t \in [0, T]$. *Let* $\boldsymbol{f}_\theta : \mathbb{R}^d \times \mathbb{R}^z \times \mathbb{R} \to \mathbb{R}^d$ *be continuous in* $t$, *uniformly Lipschitz in* $\mathbf{x}$, *and continuously differentiable in* $\mathbf{x}$. *Let* $\mathbf{x} : \mathbb{R} \to \mathbb{R}^d$ *be the unique solution for the ODE*

$$\frac{d\mathbf{x}}{dt}(t) = \boldsymbol{f}_\theta(\mathbf{x}_t, \mathbf{z}_t, t),$$

*with initial condition* $\mathbf{x}(0) = \mathbf{x}_0$. *Let* $\mathcal{L} : \mathbb{R}^d \to \mathbb{R}$ *be a scalar-valued loss function defined on the output of the neural ODE. Then* $\partial\mathcal{L}/\partial\mathbf{z}(t) := \mathbf{a}_{\mathbf{z}}(t)$ *and there exists a unique solution* $\mathbf{a}_{\mathbf{z}} : \mathbb{R} \to \mathbb{R}^z$ *to the following IVP:*

$$\mathbf{a}_{\mathbf{z}}(T) = \mathbf{0}, \qquad \frac{d\mathbf{a}_{\mathbf{z}}}{dt}(t) = -\mathbf{a}_{\mathbf{x}}(t)^\top \frac{\partial \boldsymbol{f}_\theta(\mathbf{x}_t, \mathbf{z}_t, t)}{\partial \mathbf{z}_t}.$$

## 3  Adjoint Diffusion SDEs

As recent work [30, 31] has shown, diffusion SDEs have useful properties over probability flow ODEs for image manipulation and editing. In particular, it has been shown that probability flow ODEs are invariant in Nie et al. [31, Theorem 3.2] and that diffusion SDEs are contractive in Nie et al. [31, Theorem 3.1], *i.e.*, any gap in the mismatched prior distributions $p_t(\mathbf{x}_t)$ and $\tilde{p}_t(\mathbf{x}_t)$ for the true distribution $p_t$ and edited distribution $\tilde{p}_t$ will remain between $p_0(\mathbf{x}_0)$ and $\tilde{p}_0(\mathbf{x}_0)$, whereas for diffusion SDEs the gap can be reduced between $\tilde{p}_t(\mathbf{x}_t)$ and $p_t(\mathbf{x}_t)$ as $t$ tends towards 0. Motivated by this reasoning, we present a framework for solving the adjoint diffusion SDE using exponential integrators.

The diffusion SDE with noise prediction model is given by

$$d\mathbf{x}_t = \left[ f(t)\mathbf{x}_t + \frac{g^2(t)}{\sigma_t}\boldsymbol{\epsilon}_\theta(\mathbf{x}_t, \mathbf{z}, t) \right] dt + g(t) \, d\bar{\mathbf{w}}_t, \tag{3.1}$$

where '$dt$' is an infinitesimal *negative* timestep. Note how the drift term of the SDE looks remarkably similar to the probability flow ODE sans a missing factor of $1/2$ in front of the noise prediction model. This is due to differing manipulations of the forward Kolomogorov equations—which describe the evolution of $p_t(\mathbf{x}_t)$—used by Anderson [32] to derive the reverse-time SDE and later by Song et al. [15] to derive the probability-flow ODE. This connection is *very* important as it enables one to simplify the AdjointDEIS solvers for the adjoint diffusion SDE.

We show that for the special case of Stratonovich SDEs [3] with a diffusion coefficient $\boldsymbol{g}(t)$ which does not depend on the process state $\mathbf{x}_t$, then the adjoint process has a unique strong solution that evolves with what is essentially an ODE. Intuitively, this tracks as the stochastic term $\boldsymbol{g}(t) \circ d\mathbf{w}_t$ has nothing to do with $\mathbf{x}_t$. We state this observation somewhat informally in the following theorem. The proof can be found in Appendix E.

**Theorem 3.1.** *Let* $\boldsymbol{f} : \mathbb{R}^d \times \mathbb{R} \to \mathbb{R}^d$ *be in* $\mathcal{C}_b^{\infty,1}$ *and* $\boldsymbol{g} : \mathbb{R} \to \mathbb{R}^{d \times w}$ *be in* $\mathcal{C}_b^1$. *Let* $\mathcal{L} : \mathbb{R}^d \to \mathbb{R}$ *be a scalar-valued differentiable function. Let* $\mathbf{w}_t : [0, T] \to \mathbb{R}^w$ *be a* $w$-dimensional Wiener process. Let $\mathbf{x} : [0, T] \to \mathbb{R}^d$ *solve the Stratonovich SDE*

$$d\mathbf{x}_t = \boldsymbol{f}(\mathbf{x}_t, t) \, dt + \boldsymbol{g}(t) \circ d\mathbf{w}_t,$$

*with initial condition* $\mathbf{x}_0$. *Then the adjoint process* $\mathbf{a}_{\mathbf{x}}(t) := \partial\mathcal{L}(\mathbf{x}_T)/\partial\mathbf{x}_t$ *is a strong solution to the backwards-in-time ODE*

$$d\mathbf{a}_{\mathbf{x}}(t) = -\mathbf{a}_{\mathbf{x}}(t)^\top \frac{\partial \boldsymbol{f}}{\partial \mathbf{x}_t}(\mathbf{x}_t, t) \, dt. \tag{3.2}$$

This is a boon for us, as diffusion models use only a mere scalar diffusion coefficient, $g(t)$. Therefore, the continuous adjoint equations for the diffusion SDE just simplify to an ODE. Not only that, but as mentioned before, the drift term of the diffusion SDE and probability flow ODE differ only by a factor of 2 in the term with the noise prediction network. As only the drift term of the diffusion

---

[2]French: *continue à gauche, limite à detroite*.

[3]The notation '$\circ \, d\mathbf{w}_t$' denotes Stratonovich integration which differs from integration in the Itô sense. More details on this are found in Appendix E.

SDE is used when constructing the continuous adjoint equations, it follows that the only difference between the continuous adjoint equations for the probability flow ODE and diffusion SDE is a factor of 2. Therefore, the exact solutions are given by:

**Proposition 3.1** (Exact solution of adjoint diffusion SDEs). *Given initial values* $[\mathbf{a}_\mathbf{x}(t), \mathbf{a}_\mathbf{z}(t), \mathbf{a}_\theta(t)]$ *at time* $t \in (0, T)$, *the solution* $[\mathbf{a}_\mathbf{x}(s), \mathbf{a}_\mathbf{z}(s), \mathbf{a}_\theta(s)]$ *at time* $s \in (t, T]$ *of adjoint diffusion SDEs is*

$$\mathbf{a}_\mathbf{x}(s) = \frac{\alpha_t}{\alpha_s}\mathbf{a}_\mathbf{x}(t) + \frac{2}{\alpha_s}\int_{\lambda_t}^{\lambda_s} \alpha_\lambda^2 e^{-\lambda}\mathbf{a}_\mathbf{x}(\lambda)^\top \frac{\boldsymbol{\epsilon}_\theta(\mathbf{x}_\lambda, \mathbf{z}, \lambda)}{\partial \mathbf{x}_\lambda} \, \mathrm{d}\lambda, \tag{3.3}$$

$$\mathbf{a}_\mathbf{z}(s) = \mathbf{a}_\mathbf{z}(t) + 2\int_{\lambda_t}^{\lambda_s} \alpha_\lambda e^{-\lambda}\mathbf{a}_\mathbf{x}(\lambda)^\top \frac{\partial \boldsymbol{\epsilon}_\theta(\mathbf{x}_\lambda, \mathbf{z}, \lambda)}{\partial \mathbf{z}} \, \mathrm{d}\lambda, \tag{3.4}$$

$$\mathbf{a}_\theta(s) = \mathbf{a}_\theta(t) + 2\int_{\lambda_t}^{\lambda_s} \alpha_\lambda e^{-\lambda}\mathbf{a}_\mathbf{x}(\lambda)^\top \frac{\partial \boldsymbol{\epsilon}_\theta(\mathbf{x}_\lambda, \mathbf{z}, \lambda)}{\partial \theta} \, \mathrm{d}\lambda. \tag{3.5}$$

**Remark 3.1.** *While the adjoint diffusion SDEs evolve with an ODE, the same cannot be said for the underlying state,* $\mathbf{x}_t$. *Rather this evolves with a backwards SDE (more details in Appendix E) which requires the **same** realization of the Wiener process used to sample the image as the one used in the backwards SDE.*

### 3.1 Solving Backwards Diffusion SDEs

Lu et al. [17] propose the following first-order solver for diffusion SDEs

$$\mathbf{x}_t = \frac{\alpha_t}{\alpha_s}\mathbf{x}_s - 2\sigma_t(e^h - 1)\boldsymbol{\epsilon}_\theta(\mathbf{x}_s, s) + \sigma_t\sqrt{e^{2h} - 1}\boldsymbol{\epsilon}_s, \tag{3.6}$$

where $\boldsymbol{\epsilon}_s \sim \mathcal{N}(\mathbf{0}, \mathbf{I})$. To solve the SDE backwards in time, we follow the approach initially proposed by Wu and la Torre [33] and used by later works [31]. Given a particular realization of the Wiener process that admits $\mathbf{x}_t \sim \mathcal{N}(\alpha_t \mathbf{x}_0 \mid \sigma_t^2 \mathbf{I})$, then for two samples $\mathbf{x}_t$ and $\mathbf{x}_s$ the noise $\boldsymbol{\epsilon}_s$ can be calculated by rearranging Equation (3.6) to find

$$\boldsymbol{\epsilon}_s = \frac{\mathbf{x}_t - \frac{\alpha_t}{\alpha_s}\mathbf{x}_s + 2\sigma_t(e^h - 1)\boldsymbol{\epsilon}_\theta(\mathbf{x}_s, \mathbf{z}, s)}{\sigma_t\sqrt{e^{2h} - 1}} \tag{3.7}$$

With this the sequence $\{\boldsymbol{\epsilon}_{t_i}\}_{i=1}^N$ of added noises can be calculated which will **exactly** reconstruct the original input from the initial realization of the Wiener process. This technique is referred to as *Cycle-SDE* after the CycleDiffusion paper [33].

## 4 Related Work

Our proposed solutions can be viewed as a *training-free* method for guided generation. As an active area of research, there have been several proposed approaches to the problem of training-free guided generation, which either dynamically optimize the solution trajectory during sampling [34–36], or optimize the whole solution trajectory [37–40]. Our solutions fall into the latter category of optimizing the whole solution trajectory along with additional conditional information.

While Nie et al. [37] explored the use of the continuous adjoint equations to optimize the solution trajectories of diffusion SDEs they don't consider the ODE case and make use of the *special* structure of diffusion SDEs to simplify the continuous adjoint equations as we did. Recent work by Pan et al. [39] explore the using continuous adjoint equations for guided generation but does not consider the SDE case and uses a different scheme to simplify the continuous adjoint equations. We provide a more detailed comparison against these approaches and further discussion on related methods in Appendix A.

## 5 Experiments

To illustrate the efficacy of our technique, we examine an application of guided generation in the form of the face morphing attack. The face morphing attack is a new emerging attack on Face Recognition (FR) systems. This attack works by creating a singular morphed face image $\mathbf{x}_0^{(ab)}$ that

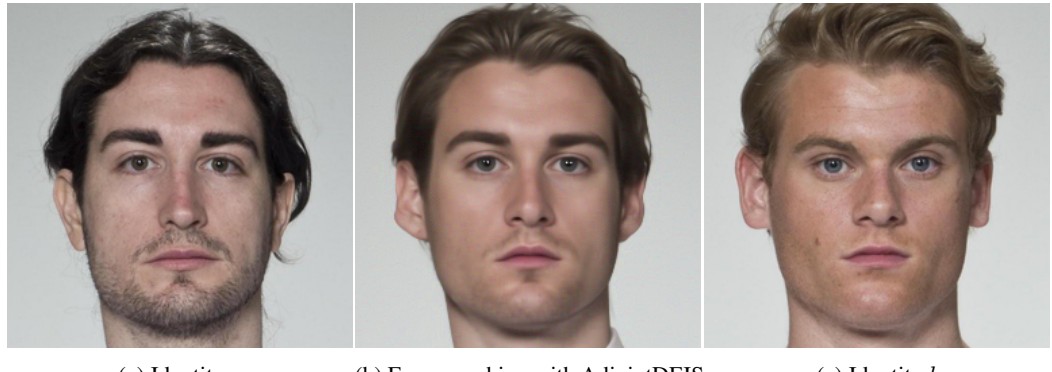

(a) Identity $a$       (b) Face morphing with AdjointDEIS       (c) Identity $b$

Figure 2: Example of guided morphed face generation with AdjointDEIS on the FRLL dataset.

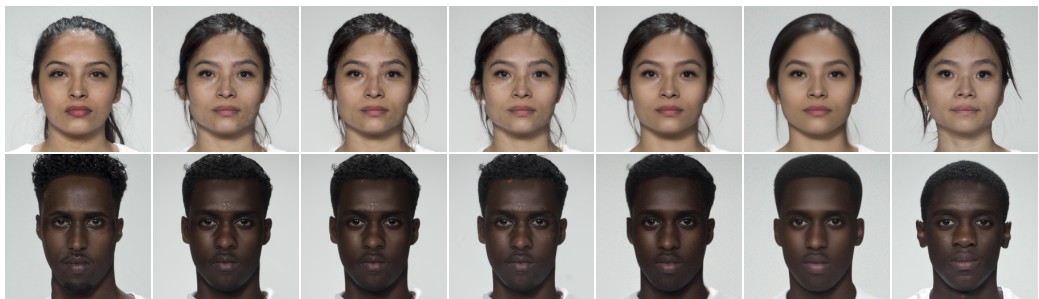

Figure 3: Comparison of DiM morphs on the FRLL dataset. From left to right, identity $a$, DiM-A, Fast-DiM, Morph-PIPE, AdjointDEIS (ODE), AdjointDEIS (SDE), and identity $b$.

shares biometric information with the two contributing faces $\mathbf{x}_0^{(a)}$ and $\mathbf{x}_0^{(b)}$ [41–43]. A successfully created morphed face image can trigger a false accept with either of the two contributing identities in the targeted Face Recognition (FR) system, see Figure 2 for an illustration. Recent work in this space has explored the use of diffusion models to generate these powerful attacks [41, 44, 45]. All prior work on diffusion-based face morphing used a pre-trained diffusion autoencoder [46] trained on the FFHQ [47] dataset at a $256 \times 256$ resolution. We illustrate the use of AdjointDEIS solvers by modifying the Diffusion Morph (DiM) architecture proposed by Blasingame and Liu [41] to use the AdjointDEIS solvers to find the optimal initial noise $\mathbf{x}_T^{(ab)}$ and conditional $\mathbf{z}_{ab}$. The AdjointDEIS solvers are used to calculate the gradients with respect to the identity loss [45] defined as

$$\mathcal{L}_{ID} = d(v_{ab}, v_a) + d(v_{ab}, v_b), \quad \mathcal{L}_{diff} = \big| d(v_{ab}, v_a) - d(v_{ab}, v_b)) \big|,$$
$$\mathcal{L}_{ID}^* = \mathcal{L}_{ID} + \mathcal{L}_{diff}, \tag{5.1}$$

where $v_a = F(\mathbf{x}_0^{(a)}), v_b = F(\mathbf{x}_0^{(b)}), v_{ab} = F(\mathbf{x}_0^{(ab)})$, and $F : \mathcal{X} \to V$ is an FR system which embeds images into a vector space $V$ which is equipped with a measure of distance, $d$. We used the ArcFace [48] FR system for identity loss.

We compare against three preexisting DiM methods, the original DiM algorithm [41], Fast-DiM [44], and Morph-PIPE [45] as well as a GAN-inversion-based face morphing attack, MIPGAN-I and MIPGAN-II [49] based on the StyleGAN [47] and StyleGAN2 [50] architectures respectively. Fast-DiM improves DiM by using higher-order ODE solvers to decrease the number of sampling steps required to create a morph. Morph-PIPE performs a very simple version of guided generation by generating a large batch of morphed images derived from a discrete set of interpolations between $\mathbf{x}_T^{(a)}$ and $\mathbf{x}_T^{(b)}$, and $\mathbf{z}_a$ and $\mathbf{z}_b$. For reference purposes, we compare against a reference GAN-based method [49] which uses GAN-inversion w.r.t.to the identity loss to find the optimal morphed face, and we include prior state-of-the-art Webmorph, a commercial off-the-shelf system [51].

We run our experiments on SYN-MAD 2022 [51] morphed pairs that are constructed from the Face Research Lab London dataset [52], more details in Appendix H.4. The morphed images are evaluated against three FR systems, the ArcFace [48], ElasticFace [53], and AdaFace [54] models;

further details are found in Appendix H.5. To measure the efficacy of a morphing attack, the Mated Morph Presentation Match Rate (MMPMR) metric [55] is used. The MMPMR metric as proposed by Scherhag et al. [55] is defined as

$$M(\delta) = \frac{1}{M} \sum_{m=1}^{M} \left\{ \left[ \min_{n \in \{1,\ldots,N_m\}} S_m^n \right] > \delta \right\}$$

(5.2)

where $\delta$ is the verification threshold, $S_m^n$ is the similarity score of the $n$-th subject of morph $m$, $N_m$ is the total number of contributing subjects to morph $m$, and $M$ is the total number of morphed images.

In our experiments, we used a learning rate of $0.01$, $N = 20$ sampling steps, $M = 20$ steps for AdjointDEIS, and 50 optimization steps for gradient descent. For the sampling process we used the DDIM solver [2], a widely used first-order solver. Following [56] we observed that using recorded values of $\{x_{t_i}\}_{i=1}^{N}$ for the backward pass improved performance. Note, this does not mean we stored the vector-Jacobians or any other internal states of the neural network. Moreover, due to our use of Cycle-SDE this choice was mandated for the SDE case. We discussion this decision further in Appendix F.1.

Table 1: Vulnerability of different FR systems across different morphing attacks on the SYN-MAD 2022 dataset. FMR = 0.1%.

| Morphing Attack | NFE($\downarrow$) | MMPMR($\uparrow$) | | |
| --- | --- | --- | --- | --- |
| | | AdaFace | ArcFace | ElasticFace |
| Webmorph [51] | - | 97.96 | 96.93 | 98.36 |
| MIPGAN-I [49] | - | 72.19 | 77.51 | 66.46 |
| MIPGAN-II [49] | - | 70.55 | 72.19 | 65.24 |
| DiM-A [41] | 350 | 92.23 | 90.18 | 93.05 |
| Fast-DiM [44] | 300 | 92.02 | 90.18 | 93.05 |
| Morph-PIPE [45] | 2350 | 95.91 | 92.84 | 95.5 |
| **DiM + AdjointDEIS-1 (ODE)** | 2250 | **99.8** | **98.77** | **99.39** |
| **DiM + AdjointDEIS-1 (SDE)** | 2250 | 98.57 | 97.96 | 97.75 |

In Table 1 we present the effectiveness of the morphing attacks against the three FR systems. Guided generation with AdjointDEIS massively increases the performance of DiM, supplanting the old state-of-the-art for face morphing. Interestingly, the SDE variant did not fare as well as the ODE variant. This is likely due to the difficulty in discretizing SDEs with large step sizes [15–17]. We present further results in Appendix F that explore the impact of the choice of learning rate and the number of discretization steps for AdjointDEIS.

# 6 Conclusion

We present a unified view on guided generation by updating latent, conditional, and model information of diffusion models with a guidance function using the continuous adjoint equations. We propose AdjointDEIS, a family of solvers for the continuous adjoint equations of diffusion models. We exploit the unique construction of diffusion models to create efficient numerical solvers by using exponential integrators. We prove the convergence order of solvers and show that the continuous adjoint equations for diffusion SDEs evolve with an ODE. Furthermore, we show how to handle conditional information that is scheduled in time, further expanding the generalizability of the proposed technique. Our results in face morphing show that the gradients produced by AdjointDEIS can be used for guided generation tasks.

**Limitations.** There are several limitations. Empirically, we only explored a small subset of the true potential AdjointDEIS by evaluating on a single scenario, *i.e.*, face morphing. Likewise, we only explored a few different hyperparameter options. In particular, we did not explore much the impact of the number of optimization steps and the number of sampling steps for diffusion SDEs on the visual quality of the generated face morphs.

**Broader Impact.** Guided generation techniques can be misused for a variety of harmful purposes. In particular, our approach provides a powerful tool for adversarial attacks. However, better knowledge of such techniques should hopefully help direct research in hardening systems against such kinds of attacks.

## Acknowledgments

The authors would like to thank Fangyikang Wang for his helpful feedback on the Taylor expansion of the vector Jacobians. The authors would also like to acknowledge the fruitful discussions with Pierre Marion and Quentin Berthet on using the adjoint methods from a bilevel optimization perspective.

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

**Organization of the appendix.** In Appendix A we provide a detailed comparison between our work and related work. Appendix B provides the full derivations for the construction of the AdjointDEIS-$k$ solvers. Likewise, in Appendix C we present the proof for Theorem 2.1. In a similar manner Appendix D presents the proof for Theorem 2.2. A more detailed discussion of the case of adjoint diffusion SDEs is held in Appendix E along with the proof of Theorem 3.1. We include additional experiments on the impact of AdjointDEIS on the face morphing problem in Appendix F. The details of the implementation of our approach are included in Appendix G. Additional details specific to our experiments are likewise included in Appendix H. Finally, for completeness, we include Appendix I to show the closed-form solutions of the drift and diffusion coefficients.

# A   Additional Related Work

In this section, we compare several recent methods for training-free guided generation which we broadly classify into two categories:

1. Techniques which directly optimize the solution trajectory during sampling [34–36]

2. Techniques which search for the optimal latents $\mathbf{x}_T$ and or $\mathbf{z}$ (this can include optimizing the solution trajectory as well) [38, 39].

In Table 2 we compare several different techniques for training-free guided diffusion in whether they explicitly optimize the *whole* solution trajectory, *i.e.*, optimizing $\mathbf{x}_T$, if they optimize additional information like $\mathbf{z}$ or $\theta$, and whether the formulation is for diffusion ODEs, SDEs, or both.

Table 2: Overview of different training-free guidance methods for diffusion models.

| Method | ODE | SDE | Optimize $\mathbf{x}_T$ | Optimize $(\mathbf{z}, \theta)$ |
|---|---|---|---|---|
| FlowGrad [36] | ✓ | ✗ | ✗ | ✗ |
| FreeDoM [34] | ✗ | ✓ | ✗ | ✗ |
| Greedy [35] | ✓ | ✓ | ✗ | ✗ |
| DOODL [38] | ✓ | ✗ | ✓ | ✗ |
| DiffPure [37] | ✗ | ✓ | ✓ | ✗ |
| AdjointDPM [39] | ✓ | ✗ | ✓ | ✓ |
| Implicit Diffusion [40] | ✓ | ✓ | ✓ | ✓ |
| **AdjointDEIS** | ✓ | ✓ | ✓ | ✓ |

Using Table 2 as a high-level overview, we spend the rest of this section providing a more detailed comparison and discussion of the related works which we separate by the two categories.

## A.1   Optimizing the Solution Trajectory During Sampling

**FlowGrad.** The FlowGrad [36] technique controls the generative process by solving the following optimal control problem

$$\min_{\boldsymbol{u}} \quad \mathcal{L}(\mathbf{x}_0) + \lambda \int_T^0 \|\boldsymbol{u}(t)\|^2 \, \mathrm{d}t, \tag{A.1}$$

$$\text{s.t.} \quad \mathbf{x}_0 = \mathbf{x}_T + \int_T^0 \boldsymbol{f}_\theta(\mathbf{x}_t, \mathbf{z}, t) + \boldsymbol{u}(t) \, \mathrm{d}t \tag{A.2}$$

where $\boldsymbol{u}$ is the control function. This optimization objective learns to alter the flow, $\boldsymbol{f}_\theta$, by $\boldsymbol{u}$. In practice, this amounts to injecting a control step governed by $\boldsymbol{u}(t)$ for a discretized schedule. This technique does not allow for learning an optimal $\mathbf{x}_T$, $\mathbf{z}$, or $\theta$.

**FreeDoM.** Another recent work, FreeDoM [34] looks at gradient guided generation of images by calculating the gradient w.r.t. $\mathbf{x}_t$ by using the approximated clean image

$$\mathbf{x}_0 \approx \frac{\mathbf{x}_t - \sigma_t \boldsymbol{\epsilon}_\theta(\mathbf{x}_t, \mathbf{z}, t)}{\alpha_t} \tag{A.3}$$

at each timestep. Let $h_i = \lambda_{t_i} - \lambda_{t_{i-1}}$. The strategy can be described as

$$\mathbf{x}_{t_{i-1}} = \frac{\alpha_{t_{i-1}}}{\alpha_{t_i}}\mathbf{x}_{t_i} - 2\sigma_{t_{i-1}}(e^{h_i} - 1)\boldsymbol{\epsilon}_\theta(\mathbf{x}_{t_i}, \mathbf{z}, t_i) + \sigma_{t_{i-1}}\sqrt{e^{2h_i} - 1}\boldsymbol{\epsilon}_{t_i}, \tag{A.4}$$

$$\hat{\mathbf{x}}_0 = \frac{\mathbf{x}_{t_i} - \sigma_{t_i}\boldsymbol{\epsilon}_\theta(\mathbf{x}_{t_i}, \mathbf{z}, t_i)}{\alpha_{t_i}}, \tag{A.5}$$

$$\mathbf{g}_{t_i} = \frac{\partial \mathcal{L}(\hat{\mathbf{x}}_0)}{\partial \mathbf{x}_t}, \tag{A.6}$$

$$\mathbf{x}_{t_{i-1}} = \mathbf{x}_{t_{i-1}} - \eta_{t_i}\mathbf{g}_{t_i}, \tag{A.7}$$

where $\eta_{t_i}$ is a learning rate defined per timestep. Importantly, FreeDoM operates on diffusion SDEs. They have an addition algorithm in which the add noise back to the image, in essence going back one timestep and applying the guidance step again.

**Greedy.** Similar to FreeDoM, greedy guided generation [35] looks to alter the generative trajectory by injecting the gradient defined on the approximated clean image; however, this technique does so w.r.t. the prediction noise, *i.e.*,

$$\boldsymbol{\epsilon}'_{t_i} = \text{stopgrad}(\boldsymbol{\epsilon}_\theta(\mathbf{x}_{t_i}, \mathbf{z}, t_i)) \tag{A.8}$$

$$\hat{\mathbf{x}}_0 = \frac{\mathbf{x}_{t_i} - \sigma_{t_i}\boldsymbol{\epsilon}'_{t_i}}{\alpha_{t_i}}, \tag{A.9}$$

$$\mathbf{g}_{t_i} = \frac{\partial \mathcal{L}(\hat{\mathbf{x}}_0)}{\partial \boldsymbol{\epsilon}_{t_i}}, \tag{A.10}$$

$$\boldsymbol{\epsilon}'_{t_i} = \boldsymbol{\epsilon}'_{t_i} - \eta_{t_i}\mathbf{g}_{t_i}. \tag{A.11}$$

The technique would work for either diffusion ODEs or SDEs.

## A.2 Optimizing the Entire Solution Trajectory

**DOODL.** The algorithm DOODL [38] looks at the gradient calculation based on the invertibility of EDICT [57]. This method can find the gradient w.r.t. $\mathbf{x}_T$; however, it cannot for the other quantities. DOODL additionally has further overhead due to the dual diffusion process of EDICT. Further analysis of DOODL compared to continuous adjoint equations for diffusion models can be found in [39].

Table 3: Comparison of solvers for the continuous adjoint equations for diffusion models.

| | DiffPure [37] | AdjointDPM [39] | **AdjointDEIS** |
|---|---|---|---|
| Discretization domain | $\epsilon_\theta$ over $t$ | $\epsilon_\theta$ over $\rho$ | $\epsilon_\theta$ over $\lambda$ |
| Solver type | Black box SDE solver | Black box ODE solver | Custom solver |
| Exponential Integrators | ✗ | ✓ | ✓ |
| Closed form SDE coefficients | ✗ | ✗ | ✓ |
| Interoperability with existing samplers | ✗ | ✗ | ✓ |
| Decoupled ODE schedule | ✗ | ✗ | ✓ |
| Supports SDEs | ✓ | ✗ | ✓ |

The remaining methods are much closer to our work as they use continuous adjoint equations for guidance. We provide a high-level summary that compares these methods to ours in Table 3.

**DiffPure.** Work by Nie et al. [37] examined the use of continuous adjoint equations for cleaning adversarial images, an application of guided generation. There exist a few key differences between their work and ours. First, while they consider the SDE case they use an Euler-Maruyama numerical scheme, meaning there is a discretization error incurred in the linear term of the drift coefficient. Moreover, the solver for the continuous adjoint equation results in an Euler scheme which suffers from a low convergence order and poor stability, particularly for stiff equations. Our approach used exponential integrators to greatly simplify the continuous adjoint equations whilst improving numerical stability by transforming the continuous adjoint equations into non-stiff form.

**AdjointDPM.** More closely related to our work is the recent AdjointDPM [39] which also explores the use of adjoint sensitivity methods for backpropagation through the *probability flow* ODE. While

they also propose to use the continuous adjoint equations to find gradients for diffusion models, our work differs in several ways which we enumerate in Table 3. For clarity, we use orange to denote their notation. They reparameterize Equation (1.4) as

$$\frac{\mathrm{d}\mathbf{y}}{\mathrm{d}\rho} = \tilde{\boldsymbol{\epsilon}}_\theta(e^{\int_0^{\gamma^{-1}(\rho)} f(\tau)\,\mathrm{d}\tau}\mathbf{y}, \gamma^{-1}(\rho), c) \tag{A.12}$$

where $c$ denotes the conditional information, $\rho = \gamma(t)$, and $\frac{\mathrm{d}\gamma}{\mathrm{d}t} = e^{-\int_0^t f(\tau)\,\mathrm{d}\tau}\frac{g^2(t)}{2\sigma_t}$. which gives them the following ODE for calculating the adjoint.

$$\frac{\mathrm{d}}{\mathrm{d}\rho}\left[\frac{\partial\mathcal{L}}{\partial\mathbf{y}_\rho}\right] = -\frac{\partial\mathcal{L}}{\partial\mathbf{y}_\rho}^\top \frac{\partial\boldsymbol{\epsilon}_\theta(e^{\int_0^{\gamma^{-1}(\rho)} f(\tau)\,\mathrm{d}\tau}\mathbf{y}_\rho, \mathbf{z}, \gamma^{-1}(\rho))}{\partial\mathbf{y}_\rho} \tag{A.13}$$

In our approach we integrate over $\lambda_t$ whereas they integrate over $\rho$. Moreover, we provide custom solvers designed specifically for diffusion ODEs instead of using a black-box ODE solver. Our approach is also interoperable with other forward ODE solvers, meaning our AdjointDEIS solver is agnostic to the ODE solver used to generate the output; however, the AdjointDPM model is tightly coupled to its forward solver. Lastly and most importantly, our method is more general and supports diffusion SDEs, not just ODEs.

Although the original paper omitted a closed form expression for $\gamma^{-1}(\rho)$, we provide to give a comparison between both methods, and to ensure that AdjointDPM can be fully implemented. In the VP SDE scheme with a linear noise schedule $\log\alpha_t$ is found to be

$$\log\alpha_t = -\frac{\beta_1 - \beta_0}{4}t^2 - \frac{\beta_0}{2}t \tag{A.14}$$

on $t \in [0, 1]$ with $\beta_0 = 0.1, \beta_1 = 20$, following Song et al. [15]. Then $\gamma^{-1}(\rho)$ is found to be

$$\gamma^{-1}(\rho) = \frac{\beta_0 - \sqrt{\beta_0^2 + 4\log\frac{1}{\sqrt{\frac{1}{\alpha_0^2}(\rho+\sigma_0)^2+1}}(\beta_0 - \beta_1)}}{\beta_0 - \beta_1} \tag{A.15}$$

**Implicit Diffusion.** Concurrent work to ours by Marion et al. [40] has also explored the use of continuous adjoint equations for the guidance of diffusion models. They, however, focus on an efficient scheme to parallelize the solution to the adjoint ODE from the perspective of bi-level optimization rather than the adjoint technique itself. So, while we focused on the numerical solvers for the continuous adjoint equations, they focused on an efficient implementation of the optimization problem from the perspective of bi-level optimization.

# B Derivation of AdjointDEIS

In this section, we provide the full derivations for the family of AdjointDEIS solvers. First recall the full definition of the continuous adjoint equations for the empirical probability flow ODE:

$$\mathbf{a}_\mathbf{x}(0) = \frac{\partial\mathcal{L}}{\partial\mathbf{x}_0}, \qquad\qquad \frac{\mathrm{d}\mathbf{a}_\mathbf{x}}{\mathrm{d}t}(t) = -\mathbf{a}_\mathbf{x}(t)^\top\frac{\partial\boldsymbol{f}_\theta(\mathbf{x}_t, \mathbf{z}, t)}{\partial\mathbf{x}_t},$$

$$\mathbf{a}_\mathbf{z}(0) = \mathbf{0}, \qquad\qquad \frac{\mathrm{d}\mathbf{a}_\mathbf{z}}{\mathrm{d}t}(t) = -\mathbf{a}_\mathbf{x}(t)^\top\frac{\partial\boldsymbol{f}_\theta(\mathbf{x}_t, \mathbf{z}, t)}{\partial\mathbf{z}},$$

$$\mathbf{a}_\theta(0) = \mathbf{0}, \qquad\qquad \frac{\mathrm{d}\mathbf{a}_\theta}{\mathrm{d}t}(t) = -\mathbf{a}_\mathbf{x}(t)^\top\frac{\partial\boldsymbol{f}_\theta(\mathbf{x}_t, \mathbf{z}, t)}{\partial\theta}. \tag{B.1}$$

We can simplify the equations by explicitly solving gradients of the neural vector field $\boldsymbol{f}_\theta$ for the drift term to obtain

$$\frac{\mathrm{d}\mathbf{a}_\mathbf{x}}{\mathrm{d}t}(t) = -f(t)\mathbf{a}_\mathbf{x}(t) - \frac{g^2(t)}{2\sigma_t}\mathbf{a}_\mathbf{x}(t)^\top\frac{\partial\boldsymbol{\epsilon}_\theta(\mathbf{x}_t, \mathbf{z}, t)}{\partial\mathbf{x}_t},$$

$$\frac{\mathrm{d}\mathbf{a}_\mathbf{z}}{\mathrm{d}t}(t) = \mathbf{0} - \frac{g^2(t)}{2\sigma_t}\mathbf{a}_\mathbf{x}(t)^\top\frac{\partial\boldsymbol{\epsilon}_\theta(\mathbf{x}_t, \mathbf{z}, t)}{\partial\mathbf{z}},$$

$$\frac{\mathrm{d}\mathbf{a}_\theta}{\mathrm{d}t}(t) = \mathbf{0} - \frac{g^2(t)}{2\sigma_t}\mathbf{a}_\mathbf{x}(t)^\top\frac{\partial\boldsymbol{\epsilon}_\theta(\mathbf{x}_t, \mathbf{z}, t)}{\partial\theta}. \tag{B.2}$$

**Remark B.1.** *The last two equations in Equations (B.2) the vector fields are independent of $(\mathbf{a}_\mathbf{z}, \mathbf{a}_\theta)$, reducing these equations to mere integrals; however, it is often useful to compute the whole system $\mathbf{a}_{aug} = (\mathbf{a}_\mathbf{x}, \mathbf{a}_\mathbf{z}, \mathbf{a}_\theta)$ as an augmented ODE.*

**Remark B.2.** *Likewise, the last two equations in Equations (B.2) are functionally identical with a simple swap of $\mathbf{z}$ for $\theta$ or vice versa.*

As such, for the sake of brevity, the derivations for the AdjointDEIS solvers for $(\mathbf{a}_\mathbf{z}, \mathbf{a}_\theta)$ will only explicitly include the derivations for $\mathbf{a}_\mathbf{z}$.

## B.1 Simplified Formulation of the Continuous Adjoint Equations

Focusing first on the continuous adjoint equation for $\mathbf{a}_\mathbf{x}$ we apply the integrating factor $\exp\left(\int_0^t f(\tau)\,\mathrm{d}\tau\right)$ to Equation (B.2) to find

$$\frac{\mathrm{d}}{\mathrm{d}t}\left[e^{\int_0^t f(\tau)\,\mathrm{d}\tau}\mathbf{a}_\mathbf{x}(t)\right] = -e^{\int_0^t f(\tau)\,\mathrm{d}\tau}\frac{g^2(t)}{2\sigma_t}\mathbf{a}_\mathbf{x}(t)^\top\frac{\partial\boldsymbol{\epsilon}_\theta(\mathbf{x}_t, \mathbf{z}, t)}{\partial\mathbf{x}_t}. \tag{B.3}$$

Then, the exact solution at time $s$ given time $t < s$ is found to be

$$e^{\int_0^s f(\tau)\,\mathrm{d}\tau}\mathbf{a}_\mathbf{x}(s) = e^{\int_0^t f(\tau)\,\mathrm{d}\tau}\mathbf{a}_\mathbf{x}(t) - \int_t^s e^{\int_0^u f(\tau)\,\mathrm{d}\tau}\frac{g^2(u)}{2\sigma_u}\mathbf{a}_\mathbf{x}(u)^\top\frac{\boldsymbol{\epsilon}_\theta(\mathbf{x}_u, \mathbf{z}, u)}{\partial\mathbf{x}_u}\,\mathrm{d}u$$

$$\mathbf{a}_\mathbf{x}(s) = e^{\int_s^t f(\tau)\,\mathrm{d}\tau}\mathbf{a}_\mathbf{x}(t) - \int_t^s e^{\int_s^u f(\tau)\,\mathrm{d}\tau}\frac{g^2(u)}{2\sigma_u}\mathbf{a}_\mathbf{x}(u)^\top\frac{\boldsymbol{\epsilon}_\theta(\mathbf{x}_u, \mathbf{z}, u)}{\partial\mathbf{x}_u}\,\mathrm{d}u \tag{B.4}$$

To simplify Equation (B.4), recall that $f(t)$ is defined as

$$f(t) = \frac{\mathrm{d}\log\alpha_t}{\mathrm{d}t}, \tag{B.5}$$

for VP type SDEs. Furthermore, let $\lambda_t := \log(\alpha_t/\sigma_t)$ be one half of the log-SNR. Then the diffusion coefficient can be simplified using the log-derivative trick such that

$$g^2(t) = \frac{\mathrm{d}\sigma_t^2}{\mathrm{d}t} - 2\frac{\mathrm{d}\log\alpha_t}{\mathrm{d}t}\sigma_t^2 = 2\sigma_t^2\left(\frac{\mathrm{d}\log\sigma_t}{\mathrm{d}t} - \frac{\mathrm{d}\log\alpha_t}{\mathrm{d}t}\right) = -2\sigma_t^2\frac{\mathrm{d}\lambda_t}{\mathrm{d}t}. \tag{B.6}$$

Using this updated expression of $g^2(t)$ along with computing the integrating factor in closed form enables us to express Equation (B.4) as

$$\mathbf{a}_\mathbf{x}(s) = \frac{\alpha_t}{\alpha_s}\mathbf{a}_\mathbf{x}(t) + \frac{1}{\alpha_s}\int_t^s \alpha_u\sigma_u\frac{\mathrm{d}\lambda_u}{\mathrm{d}u}\mathbf{a}_\mathbf{x}(u)^\top\frac{\boldsymbol{\epsilon}_\theta(\mathbf{x}_u, \mathbf{z}, u)}{\partial\mathbf{x}_u}\,\mathrm{d}u. \tag{B.7}$$

Lastly, by rewriting the integral in terms of an exponentially weighted integral $\alpha_u\sigma_u = \alpha_u^2\sigma_u/\alpha_u = \alpha_u^2 e^{-\lambda_u}$ we find

$$\mathbf{a}_\mathbf{x}(s) = \frac{\alpha_t}{\alpha_s}\mathbf{a}_\mathbf{x}(t) + \frac{1}{\alpha_s}\int_{\lambda_t}^{\lambda_s} \alpha_\lambda^2 e^{-\lambda}\mathbf{a}_\mathbf{x}(\lambda)^\top\frac{\boldsymbol{\epsilon}_\theta(\mathbf{x}_\lambda, \mathbf{z}, \lambda)}{\partial\mathbf{x}_\lambda}\,\mathrm{d}\lambda. \tag{B.8}$$

This change of variables is possible as $\lambda_t$ is a strictly decreasing function w.r.t. $t$ and therefore it has an inverse function $t_\lambda$ which satisfies $t_\lambda(\lambda_t) = t$, and, with abuse of notation, we let $\mathbf{x}_\lambda := \mathbf{x}_{t_\lambda(\lambda)}$, $\mathbf{a}_\mathbf{x}(\lambda) := \mathbf{a}_\mathbf{x}(t_\lambda(\lambda))$, &c.and let the reader infer from context if the function is mapping the log-SNR back into the time domain or already in the time domain.

Now we will show the derivations to find a simplified form of the continuous adjoint equation for the conditional information. Using the continuous adjoint equation from Equations (B.2) for $\mathbf{a}_\mathbf{z}(t)$ along with the log-SNR, we can express the evolution of $\mathbf{a}_\mathbf{z}(t)$ as

$$\frac{\mathrm{d}\mathbf{a}_\mathbf{z}}{\mathrm{d}t}(t) = \sigma_t\frac{\mathrm{d}\lambda_t}{\mathrm{d}t}\mathbf{a}_\mathbf{x}(t)^\top\frac{\partial\boldsymbol{\epsilon}_\theta(\mathbf{x}_t, \mathbf{z}, t)}{\partial\mathbf{z}}. \tag{B.9}$$

As we would like to express this as an exponential integrator, we simply multiply $\sigma_t$ by $\alpha_t/\alpha_t$ to obtain $\alpha_t \cdot \sigma_t/\alpha_t = \alpha_t e^{-\lambda_t}$, as such we can rewrite Equation (B.9) as

$$\frac{\mathrm{d}\mathbf{a}_\mathbf{z}}{\mathrm{d}t}(t) = \alpha_t e^{-\lambda_t}\frac{\mathrm{d}\lambda_t}{\mathrm{d}t}\mathbf{a}_\mathbf{x}(t)^\top\frac{\partial\boldsymbol{\epsilon}_\theta(\mathbf{x}_t, \mathbf{z}, t)}{\partial\mathbf{z}}. \tag{B.10}$$

Using Equations (B.8) and (B.10), we arrive at Proposition 2.1 from the main paper.

**Proposition B.1.** *Given initial values* $[\mathbf{a_x}(t), \mathbf{a_z}(t), \mathbf{a_\theta}(t)]$ *at time* $t \in (0, T)$, *the solution* $[\mathbf{a_x}(s), \mathbf{a_z}(s), \mathbf{a_\theta}(s)]$ *at time* $s \in (t, T]$ *of the adjoint empirical probability flow ODE in Equation* (B.4) *is*

$$\mathbf{a_x}(s) = \frac{\alpha_t}{\alpha_s}\mathbf{a_x}(t) + \frac{1}{\alpha_s}\int_{\lambda_t}^{\lambda_s}\alpha_\lambda^2 e^{-\lambda}\mathbf{a_x}(\lambda)^\top \frac{\partial \boldsymbol{\epsilon}_\theta(\mathbf{x}_\lambda, \mathbf{z}, \lambda)}{\partial \mathbf{x}_\lambda}\, \mathrm{d}\lambda, \tag{B.11}$$

$$\mathbf{a_z}(s) = \mathbf{a_z}(t) + \int_{\lambda_t}^{\lambda_s}\alpha_\lambda e^{-\lambda}\mathbf{a_x}(\lambda)^\top \frac{\partial \boldsymbol{\epsilon}_\theta(\mathbf{x}_\lambda, \mathbf{z}, \lambda)}{\partial \mathbf{z}}\, \mathrm{d}\lambda, \tag{B.12}$$

$$\mathbf{a_\theta}(s) = \mathbf{a_\theta}(t) + \int_{\lambda_t}^{\lambda_s}\alpha_\lambda e^{-\lambda}\mathbf{a_x}(\lambda)^\top \frac{\partial \boldsymbol{\epsilon}_\theta(\mathbf{x}_\lambda, \mathbf{z}, \lambda)}{\partial \theta}\, \mathrm{d}\lambda. \tag{B.13}$$

Then to find the AdjointDEIS solvers we take a $k$-th order Taylor expansion about $\lambda_t$ and integrate in the log-SNR domain.

## B.2 Taylor Expansion

For $k \geq 1$, the $(k-1)$-th Taylor expansion at $\lambda_t$ of the inner term of the exponentially weighted integral in Equation (B.11) is

$$\mathbf{a_x}(\lambda)^\top \frac{\partial \boldsymbol{\epsilon}_\theta(\mathbf{x}_\lambda, \mathbf{z}, \lambda)}{\partial \mathbf{x}_\lambda} = \sum_{n=0}^{k-1}\frac{(\lambda - \lambda_t)^n}{n!}\frac{\mathrm{d}^n}{\mathrm{d}\lambda^n}\left[\alpha_\lambda^2 \mathbf{a_x}(\lambda)^\top \frac{\partial \boldsymbol{\epsilon}_\theta(\mathbf{x}_\lambda, \mathbf{z}, \lambda)}{\partial \mathbf{x}_\lambda}\right]_{\lambda=\lambda_t} + \mathcal{O}((\lambda - \lambda_t)^k). \tag{B.14}$$

Then plugging this into Equation (B.11) yields

$$
\begin{aligned}
\mathbf{a_x}(s) &= \frac{\alpha_t}{\alpha_s}\mathbf{a_x}(t) \quad + \frac{1}{\alpha_s}\int_{\lambda_t}^{\lambda_s}e^{-\lambda}\sum_{n=0}^{k-1}\frac{(\lambda - \lambda_t)^n}{n!}\frac{\mathrm{d}^n}{\mathrm{d}\lambda^n}\left[\alpha_\lambda^2 \mathbf{a_x}(\lambda)^\top \frac{\partial \boldsymbol{\epsilon}_\theta(\mathbf{x}_\lambda, \mathbf{z}, \lambda)}{\partial \mathbf{x}_\lambda}\right]_{\lambda=\lambda_t}\mathrm{d}\lambda \\
&\quad + \mathcal{O}(h^{k+1}) \\
&= \frac{\alpha_t}{\alpha_s}\mathbf{a_x}(t) \quad + \frac{1}{\alpha_s}\sum_{n=0}^{k-1}\underbrace{\frac{\mathrm{d}^n}{\mathrm{d}\lambda^n}\left[\alpha_\lambda^2 \mathbf{a_x}(\lambda)^\top \frac{\partial \boldsymbol{\epsilon}_\theta(\mathbf{x}_\lambda, \mathbf{z}, \lambda)}{\partial \mathbf{x}_\lambda}\right]_{\lambda=\lambda_t}}_{\text{estimated}}\underbrace{\int_{\lambda_t}^{\lambda_s}\frac{(\lambda - \lambda_t)^n}{n!}e^{-\lambda}\,\mathrm{d}\lambda}_{\text{analytically computed}} \\
&\quad + \underbrace{\mathcal{O}(h^{k+1})}_{\text{omitted}},
\end{aligned}
\tag{B.15}
$$

where $h = \lambda_s - \lambda_t$.

The exponentially weighted integral $\int_{\lambda_t}^{\lambda_s}\frac{(\lambda - \lambda_t)^n}{n!}e^{-\lambda}\,\mathrm{d}\lambda$ can be solved *analytically* by applying $n$ times integration by parts [16, 26] such that

$$\int_{\lambda_t}^{\lambda_s}e^{-\lambda}\frac{(\lambda - \lambda_t)^n}{n!}\,\mathrm{d}\lambda = \frac{\sigma_s}{\alpha_s}h^{n+1}\varphi_{n+1}(h), \tag{B.16}$$

with the special $\varphi$-functions [26]. These functions are defined as

$$\varphi_{n+1}(h) := \int_0^1 e^{(1-u)h}\frac{u^n}{n!}\,\mathrm{d}u, \qquad \varphi_0(h) = e^h, \tag{B.17}$$

which satisfy the recurrence relation $\varphi_{k+1}(h) = (\varphi_k(h) - \varphi_k(0))/h$ and have closed forms for $k = 1, 2$:

$$\varphi_1(h) = \frac{e^h - 1}{h}, \tag{B.18}$$

$$\varphi_2(h) = \frac{e^h - h - 1}{h^2}. \tag{B.19}$$

Likewise, the Taylor expansion of the exponentially weighted integral in Equation (B.12) yields

$$
\mathbf{a_z}(s) = \mathbf{a_z}(t) + \int_{\lambda_t}^{\lambda_s} e^{-\lambda} \sum_{n=0}^{k-1} \frac{(\lambda - \lambda_t)^n}{n!} \frac{\mathrm{d}^n}{\mathrm{d}\lambda^n} \left[ \alpha_\lambda \mathbf{a_x}(\lambda)^\top \frac{\partial \boldsymbol{\epsilon}_\theta(\mathbf{x}_\lambda, \mathbf{z}, \lambda)}{\partial \mathbf{z}} \right]_{\lambda=\lambda_t} \mathrm{d}\lambda + \mathcal{O}(h^{k+1})
$$

$$
= \mathbf{a_z}(t) + \sum_{n=0}^{k-1} \underbrace{\frac{\mathrm{d}^n}{\mathrm{d}\lambda^n} \left[ \alpha_\lambda \mathbf{a_x}(\lambda)^\top \frac{\partial \boldsymbol{\epsilon}_\theta(\mathbf{x}_\lambda, \mathbf{z}, \lambda)}{\partial \mathbf{z}} \right]_{\lambda=\lambda_t}}_{\text{estimated}} \underbrace{\int_{\lambda_t}^{\lambda_s} \frac{(\lambda - \lambda_t)^n}{n!} e^{-\lambda} \, \mathrm{d}\lambda}_{\text{analytically computed}} + \underbrace{\mathcal{O}(h^{k+1})}_{\text{omitted}}.
$$

(B.20)

## B.3  AdjointDEIS-1

For $k = 1$ and omitting the higher-order error term, Equation (B.15) becomes:

$$
\mathbf{a_x}(s) = \frac{\alpha_t}{\alpha_s} \mathbf{a_x}(t) + \frac{1}{\alpha_s} \alpha_t^2 \mathbf{a_x}(t)^\top \frac{\partial \boldsymbol{\epsilon}_\theta(\mathbf{x}_t, \mathbf{z}, t)}{\partial \mathbf{x}_t} \int_{\lambda_t}^{\lambda_s} \frac{(\lambda - \lambda_t)^0}{0!} e^{-\lambda} \, \mathrm{d}\lambda
$$

$$
= \frac{\alpha_t}{\alpha_s} \mathbf{a_x}(t) + \sigma_s(e^h - 1) \frac{\alpha_t^2}{\alpha_s^2} \mathbf{a_x}(t)^\top \frac{\partial \boldsymbol{\epsilon}_\theta(\mathbf{x}_t, \mathbf{z}, t)}{\partial \mathbf{x}_t} \qquad \text{By Equation (B.16).} \qquad \text{(B.21)}
$$

Likewise, the continuous adjoint equation for $\mathbf{z}$, Equation (B.20), becomes when $k = 1$ by omitting the higher-order error term:

$$
\mathbf{a_z}(s) = \mathbf{a_z}(t) + \alpha_t \mathbf{a_x}(t)^\top \frac{\partial \boldsymbol{\epsilon}_\theta(\mathbf{x}_t, \mathbf{z}, t)}{\partial \mathbf{z}} \int_{\lambda_t}^{\lambda_s} \frac{(\lambda - \lambda_t)^0}{0!} e^{-\lambda} \, \mathrm{d}\lambda
$$

$$
= \mathbf{a_z}(t) + \sigma_s(e^h - 1) \frac{\alpha_t}{\alpha_s} \mathbf{a_x}(t)^\top \frac{\partial \boldsymbol{\epsilon}_\theta(\mathbf{x}_t, \mathbf{z}, t)}{\partial \mathbf{z}} \qquad \text{By Equation (B.16).} \qquad \text{(B.22)}
$$

And the first-order solver for $\mathbf{a}_\theta(t)$ can be found in a similar fashion, thus we have derived the AdjointDEIS-1 solvers.

## B.4  AdjointDEIS-2M

Consider the following definition of the limit in the log-SNR domain

$$
\frac{\mathrm{d}}{\mathrm{d}\lambda} \left[ \alpha_\lambda^2 \mathbf{a_x}(\lambda)^\top \frac{\partial \boldsymbol{\epsilon}_\theta(\mathbf{x}_\lambda, \mathbf{z}, \lambda)}{\partial \mathbf{x}_\lambda} \right] = \lim_{\lambda_r \to \lambda_t} \frac{\mathbf{V}(\mathbf{x}; \lambda_t) - \mathbf{V}(\mathbf{x}; \lambda_r)}{\rho h}, \qquad \text{(B.23)}
$$

where $\rho = \frac{\lambda_t - \lambda_r}{h}$ with $h = \lambda_s - \lambda_t$ and where $r$ is some previous step $r < t < s$. Again, $\mathbf{V}(\mathbf{x}; \lambda_t)$ is overloaded to mean $\mathbf{V}(\mathbf{x}; t_\lambda(\lambda_t))$. Then by omitting higher-order error $\mathcal{O}(h^{k+1})$, Equation (B.15) becomes:

$$
\mathbf{a_x}(s) = \frac{\alpha_t}{\alpha_s} \mathbf{a_x}(t) + \frac{1}{\alpha_s} \left[ \mathbf{V}(\mathbf{x}; \lambda_t) \int_{\lambda_t}^{\lambda_s} \frac{(\lambda - \lambda_t)^0}{0!} \, \mathrm{d}\lambda + \mathbf{V}^{(1)}(\mathbf{x}; \lambda_t) \int_{\lambda_t}^{\lambda_s} \frac{(\lambda - \lambda_t)^1}{1!} \, \mathrm{d}\lambda \right]
$$

$$
= \frac{\alpha_t}{\alpha_s} \mathbf{a_x}(t) + \frac{1}{\alpha_s} \left[ \frac{\sigma_s}{\alpha_s}(e^h - 1) \mathbf{V}(\mathbf{x}; \lambda_t) + \frac{\sigma_s}{\alpha_s}(e^h - h - 1) \mathbf{V}^{(1)}(\mathbf{x}; \lambda_t) \right]. \qquad \text{(B.24)}
$$

By applying the same approximation used in Lu et al. [16] of

$$
\frac{e^h - h - 1}{h} \approx \frac{e^h - 1}{2}, \qquad \text{(B.25)}
$$

then we can rewrite the second term of the Taylor expansion as

$$
\frac{\sigma_s}{\alpha_s}(e^h - h - 1) \mathbf{V}^{(1)}(\mathbf{x}; \lambda_t) \approx \frac{\sigma_s}{\alpha_s}(e^h - h - 1) \frac{\mathbf{V}(\mathbf{x}; \lambda_t) - \mathbf{V}(\mathbf{x}; \lambda_r)}{\rho h} \qquad \text{By Equation (B.23)}
$$

$$
\approx \frac{\sigma_s}{\alpha_s} \frac{e^h - 1}{2\rho} \left( \mathbf{V}(\mathbf{x}; \lambda_t) - \mathbf{V}(\mathbf{x}; \lambda_r) \right) \qquad \text{By Equation (B.25)}
$$

$$
= \frac{\sigma_s}{\alpha_s} \frac{e^h - 1}{2\rho} \left( \alpha_t^2 \mathbf{a_x}(t)^\top \frac{\partial \boldsymbol{\epsilon}_\theta(\mathbf{x}_t, \mathbf{z}, t)}{\partial \mathbf{x}_t} - \alpha_r^2 \mathbf{a_x}(r)^\top \frac{\partial \boldsymbol{\epsilon}_\theta(\mathbf{x}_r, \mathbf{z}, r)}{\partial \mathbf{x}_r} \right).
$$

(B.26)

Then Equation (B.24) becomes

$$\mathbf{a_x}(s) = \frac{\alpha_t}{\alpha_s}\mathbf{a_x}(t) + \sigma_s(e^h - 1)\frac{\alpha_t^2}{\alpha_s^2}\mathbf{a_x}(t)^\top\frac{\partial\boldsymbol{\epsilon}_\theta(\mathbf{x}_t, \mathbf{z}, t)}{\partial\mathbf{x}_t}$$
$$+ \sigma_s\frac{e^h - 1}{2\rho}\left(\frac{\alpha_t^2}{\alpha_s^2}\mathbf{a_x}(t)^\top\frac{\partial\boldsymbol{\epsilon}_\theta(\mathbf{x}_t, \mathbf{z}, t)}{\partial\mathbf{x}_t} - \frac{\alpha_r^2}{\alpha_s^2}\mathbf{a_x}(r)^\top\frac{\partial\boldsymbol{\epsilon}_\theta(\mathbf{x}_r, \mathbf{z}, r)}{\partial\mathbf{x}_r}\right). \quad \text{(B.27)}$$

Likewise, consider the scaled vector-Jacobian product of the adjoint state $\mathbf{a_x}(t)$ and the gradient of the model w.r.t. $\mathbf{z}$, *i.e.*,

$$\mathbf{V}(\mathbf{z};t) = \alpha_t\mathbf{a_x}(t)^\top\frac{\partial\boldsymbol{\epsilon}_\theta(\mathbf{x}_t, \mathbf{z}, t)}{\partial\mathbf{z}}, \quad \text{(B.28)}$$

along with a corresponding definition of first-derivative w.r.t. $\lambda$ as defined in Equation (B.23). As such Equation (B.20), when $k = 2$, becomes the following when omitting the higher-order error term:

$$\mathbf{a_z}(s) = \mathbf{a_z}(t) + \mathbf{V}(\mathbf{z};\lambda_t)\int_{\lambda_t}^{\lambda_s}\frac{(\lambda - \lambda_t)^0}{0!}\,\mathrm{d}\lambda + \mathbf{V}^{(1)}(\mathbf{z};\lambda_t)\int_{\lambda_t}^{\lambda_s}\frac{(\lambda - \lambda_t)^1}{1!}\,\mathrm{d}\lambda$$
$$= \mathbf{a_z}(t) + \frac{\sigma_s}{\alpha_s}(e^h - 1)\mathbf{V}(\mathbf{z};\lambda_t) + \frac{\sigma_s}{\alpha_s}(e^h - h - 1)\mathbf{V}^{(1)}(\mathbf{z};\lambda_t). \quad \text{(B.29)}$$

The second term of the Taylor expansion can be rewritten as

$$\frac{\sigma_s}{\alpha_s}(e^h - h - 1)\mathbf{V}^{(1)}(\mathbf{z};\lambda_t) \approx \frac{\sigma_s}{\alpha_s}(e^h - h - 1)\frac{\mathbf{V}(\mathbf{z};\lambda_t) - \mathbf{V}(\mathbf{z};\lambda_r)}{\rho h}$$
$$\approx \frac{\sigma_s}{\alpha_s}\frac{e^h - 1}{2\rho}\big(\mathbf{V}(\mathbf{z};\lambda_t) - \mathbf{V}(\mathbf{z};\lambda_r)\big) \qquad \text{By Equation (B.25)}$$
$$= \frac{\sigma_s}{\alpha_s}\frac{e^h - 1}{2\rho}\left(\alpha_t\mathbf{a_x}(t)^\top\frac{\partial\boldsymbol{\epsilon}_\theta(\mathbf{x}_t, \mathbf{z}, t)}{\partial\mathbf{z}} - \alpha_r\mathbf{a_x}(r)^\top\frac{\partial\boldsymbol{\epsilon}_\theta(\mathbf{x}_r, \mathbf{z}, r)}{\partial\mathbf{z}}\right). \quad \text{(B.30)}$$

Then Equation (B.29) becomes

$$\mathbf{a_z}(s) = \mathbf{a_z}(t) + \sigma_s(e^h - 1)\frac{\alpha_t}{\alpha_s}\mathbf{a_x}(t)^\top\frac{\partial\boldsymbol{\epsilon}_\theta(\mathbf{x}_t, \mathbf{z}, t)}{\partial\mathbf{z}}$$
$$+ \sigma_s\frac{e^h - 1}{2\rho}\left(\frac{\alpha_t}{\alpha_s}\mathbf{a_x}(t)^\top\frac{\partial\boldsymbol{\epsilon}_\theta(\mathbf{x}_t, \mathbf{z}, t)}{\partial\mathbf{z}} - \frac{\alpha_r}{\alpha_s}\mathbf{a_x}(r)^\top\frac{\partial\boldsymbol{\epsilon}_\theta(\mathbf{x}_r, \mathbf{z}, r)}{\partial\mathbf{z}}\right), \quad \text{(B.31)}$$

and the corresponding second-order solver for $\mathbf{a}_\theta(t)$ can be found in a similar manner.

# C  Proof of Theorem 2.1

For notational brevity we denote the *scaled* vector-Jacobian products of the solution trajectory of AdjointDEIS as

$$\tilde{\mathbf{V}}(\mathbf{x};t) = \alpha_t^2\tilde{\mathbf{a}}_\mathbf{x}(t)^\top\frac{\partial\boldsymbol{\epsilon}_\theta(\tilde{\mathbf{x}}_t, \mathbf{z}, t)}{\partial\tilde{\mathbf{x}}_t}, \quad \text{(C.1)}$$

$$\tilde{\mathbf{V}}(\mathbf{z};t) = \alpha_t\tilde{\mathbf{a}}_\mathbf{x}(t)^\top\frac{\partial\boldsymbol{\epsilon}_\theta(\tilde{\mathbf{x}}_t, \mathbf{z}, t)}{\partial\mathbf{z}}. \quad \text{(C.2)}$$

## C.1  Assumptions

For the AdjointDEIS solvers, we make similar assumptions to Lu et al. [16].

**Assumption C.1.** *The total derivatives of the vector-Jacobian products $\mathbf{V}^{(n)}(\{\mathbf{x}_\lambda, \mathbf{z}, \theta\}, \lambda)$ as a function of $\lambda$ exist and are continuous for $0 \leq n \leq k - 1$ (and hence bounded).*

**Assumption C.2.** *The function $\boldsymbol{\epsilon}_\theta(\mathbf{x}, \mathbf{z}, t)$ is continuous in $t$ and uniformly Lipschitz and continuously differentiable w.r.t. its first parameter $\mathbf{x}$.*

**Assumption C.3.** $h_{max} := \max_{1 \leq j \leq M} h_j = \mathcal{O}(1/M)$.

**Assumption C.4.** $\rho_i > c > 0$ *for all* $i = 1, \ldots, M$ *and some constant c.*

The first assumption is required by Taylor's theorem. The second assumption is a mild assumption to ensure that Theorem C.1 holds, which is used to replace $\tilde{\mathbf{V}}(\{\mathbf{x}_t, \mathbf{z}, \theta\}, t)$ with $\mathbf{V}(\{\mathbf{x}_t, \mathbf{z}, \theta\}, t) + \mathcal{O}(\tilde{\mathbf{a}}_{\mathbf{x}}(t) - \mathbf{a}_{\mathbf{x}}(t))$ so the Taylor expansion w.r.t. $\lambda_s$ is applicable. The third assumption is a technical assumption to exclude a significantly large step size. The last assumption is necessary for the case when $k = 2$. For our proofs, we follow a similar outline to that taken by Lu et al. [17, Appendix A].

## C.2 The Vector-Jacobian Product is Lipschitz

**Lemma C.1** (Vector-Jacobian Product is Lipschitz.)**.** *Let* $\boldsymbol{f}_\theta : \mathbb{R}^d \times \mathbb{R}^z \times [0, T] \to \mathbb{R}^d$ *be continuous in t and uniformly Lipschitz and continuously differentiable in* $\mathbf{x}$*. Let* $\mathbf{x} : [0, T] \to \mathbb{R}^d$ *be the unique solution to*

$$\frac{\mathrm{d}\mathbf{x}_t}{\mathrm{d}t} = \boldsymbol{f}_\theta(\mathbf{x}_t, \mathbf{z}, t)$$

*with initial condition* $\mathbf{x}_0$*. Then the following map*

$$(\mathbf{a}, t) \mapsto -\mathbf{a}^\top \frac{\partial \boldsymbol{f}_\theta(\mathbf{x}_t, \mathbf{z}, t)}{\partial[\mathbf{x}_t, \mathbf{z}, \theta]}$$

*is Lipschitz in* $\mathbf{a}$*. Moreover, the Lipschitz constant* $L > 0$ *is given by*

$$L = \sup_{t \in [0,T]} \left| \frac{\partial \boldsymbol{f}_\theta(\mathbf{x}_t, \mathbf{z}, t)}{\partial \mathbf{x}_t} \right|. \tag{C.3}$$

*Proof.* Now, as $\mathbf{x}_t$ is continuous and $\boldsymbol{f}_\theta$ is continuously differentiable in $\mathbf{x}$, so $t \mapsto \frac{\partial \boldsymbol{f}_\theta}{\partial[\mathbf{x}_t, \mathbf{z}, \theta]}(\mathbf{x}_t, \mathbf{z}, t)$ is a continuous function on the compact set $[0, T]$, so it is bounded by some $L > 0$. Likewise, for $\mathbf{a} \in \mathbb{R}^d$ the map $(\mathbf{a}, t) \mapsto -\mathbf{a}^\top \frac{\partial \boldsymbol{f}_\theta(\mathbf{x}_t, \mathbf{z}, t)}{\partial[\mathbf{x}_t, \mathbf{z}, \theta]}$ is Lipschitz in $\mathbf{a}$ with Lipschitz constant $L$ and this constant is independent of $t$. $\square$

## C.3 Proof of Theorem 2.1 when $k = 1$

*Proof.* First, we consider the case of the adjoint state $\mathbf{a}_{\mathbf{x}}(t)$. Recall that the AdjointDEIS-1 solver for $\mathbf{a}_{\mathbf{x}}$ with higher-order error terms is given by

$$\mathbf{a}_{\mathbf{x}}(t_{i+1}) = \frac{\alpha_{t_i}}{\alpha_{t_{i+1}}} \mathbf{a}_{\mathbf{x}}(t_i) + \sigma_{t_{i+1}}(e^{h_i} - 1) \frac{\alpha_{t_i}^2}{\alpha_{t_{i+1}}^2} \mathbf{a}_{\mathbf{x}}(t_i)^\top \frac{\partial \boldsymbol{\epsilon}_\theta(\mathbf{x}_{t_i}, \mathbf{z}, t)}{\partial \mathbf{x}_{t_i}} + \mathcal{O}(h_i^2), \tag{C.4}$$

where we let $t_i = t$, $t_{i+1} = s$, $h_i = \lambda_{t_{i+1}} - \lambda_{t_i}$ from Equation (B.21). By Theorem C.1 and Equation (B.21) it holds that

$$\begin{aligned}
\tilde{\mathbf{a}}_{\mathbf{x}}(t_{i+1}) &= \frac{\alpha_{t_i}}{\alpha_{t_{i+1}}} \tilde{\mathbf{a}}_{\mathbf{x}}(t_i) + \sigma_{t_{i+1}}(e^{h_i} - 1) \frac{\alpha_{t_i}^2}{\alpha_{t_{i+1}}^2} \tilde{\mathbf{a}}_{\mathbf{x}}(t_i)^\top \frac{\partial \boldsymbol{\epsilon}_\theta(\tilde{\mathbf{x}}_{t_i}, \mathbf{z}, t)}{\partial \tilde{\mathbf{x}}_{t_i}} \\
&= \frac{\alpha_{t_i}}{\alpha_{t_{i+1}}} \tilde{\mathbf{a}}_{\mathbf{x}}(t_i) + \sigma_{t_{i+1}}(e^{h_i} - 1) \frac{\alpha_{t_i}^2}{\alpha_{t_{i+1}}^2} \left( \mathbf{a}_{\mathbf{x}}(t_i)^\top \frac{\partial \boldsymbol{\epsilon}_\theta(\mathbf{x}_{t_i}, \mathbf{z}, t)}{\partial \mathbf{x}_{t_i}} + \mathcal{O}(\tilde{\mathbf{a}}_{\mathbf{x}}(t_i) - \mathbf{a}_{\mathbf{x}}(t_i)) \right) \\
&= \frac{\alpha_{t_i}}{\alpha_{t_{i+1}}} \mathbf{a}_{\mathbf{x}}(t_i) + \sigma_{t_{i+1}}(e^{h_i} - 1) \frac{\alpha_{t_i}^2}{\alpha_{t_{i+1}}^2} \mathbf{a}_{\mathbf{x}}(t_i)^\top \frac{\partial \boldsymbol{\epsilon}_\theta(\mathbf{x}_{t_i}, \mathbf{z}, t)}{\partial \mathbf{x}_{t_i}} + \mathcal{O}(\tilde{\mathbf{a}}_{\mathbf{x}}(t_i) - \mathbf{a}_{\mathbf{x}}(t_i)) \\
&= \mathbf{a}_{\mathbf{x}}(t_{i+1}) + \mathcal{O}(h_{max}^2) + \mathcal{O}(\tilde{\mathbf{a}}_{\mathbf{x}}(t_i) - \mathbf{a}_{\mathbf{x}}(t_i)). \tag{C.5}
\end{aligned}$$

Repeat, this argument, from $\tilde{\mathbf{a}}_{\mathbf{x}}(t_0) = \mathbf{a}_{\mathbf{x}}(0)$ then we find

$$\tilde{\mathbf{a}}_{\mathbf{x}}(t_M) = \mathbf{a}_{\mathbf{x}}(T) + \mathcal{O}(Mh_{max}^2) = \mathbf{a}_{\mathbf{x}}(T) + \mathcal{O}(h_{max}). \tag{C.6}$$

Although the argument for the adjoint state $\mathbf{a}_{\mathbf{z}}(t)$ follows an analogous form to the one above, we explicitly state it for completeness. Recall that the AdjointDEIS-1 solver for $\mathbf{a}_{\mathbf{z}}$ with higher-order error terms is given by

$$\mathbf{a}_{\mathbf{z}}(t_{i+1}) = \mathbf{a}_{\mathbf{z}}(t_i) + \sigma_{t_{i+1}}(e^{h_i} - 1) \frac{\alpha_{t_i}}{\alpha_{t_{i+1}}} \mathbf{a}_{\mathbf{x}}(t_i)^\top \frac{\partial \boldsymbol{\epsilon}_\theta(\mathbf{x}_{t_i}, \mathbf{z}, t)}{\partial \mathbf{z}} + \mathcal{O}(h_i^2). \tag{C.7}$$

By Theorem C.1 and Equation (B.22) it holds that

$$
\begin{aligned}
\tilde{\mathbf{a}}_{\mathbf{z}}(t_{i+1}) &= \tilde{\mathbf{a}}_{\mathbf{z}}(t_i) + \sigma_{t_{i+1}}(e^{h_i}-1)\frac{\alpha_{t_i}}{\alpha_{t_{i+1}}}\tilde{\mathbf{a}}_{\mathbf{x}}(t_i)^{\top}\frac{\partial \boldsymbol{\epsilon}_{\theta}(\tilde{\mathbf{x}}_{t_i},\mathbf{z},t)}{\partial \mathbf{z}} \\
&= \tilde{\mathbf{a}}_{\mathbf{z}}(t_i) + \sigma_{t_{i+1}}(e^{h_i}-1)\frac{\alpha_{t_i}}{\alpha_{t_{i+1}}}\left(\mathbf{a}_{\mathbf{x}}(t_i)^{\top}\frac{\partial \boldsymbol{\epsilon}_{\theta}(\mathbf{x}_{t_i},\mathbf{z},t)}{\partial \mathbf{z}} + \mathcal{O}(\tilde{\mathbf{a}}_{\mathbf{x}}(t_i)-\mathbf{a}_{\mathbf{x}}(t_i))\right) \\
&= \mathbf{a}_{\mathbf{z}}(t_i) + \sigma_{t_{i+1}}(e^{h_i}-1)\frac{\alpha_{t_i}}{\alpha_{t_{i+1}}}\mathbf{a}_{\mathbf{x}}(t_i)^{\top}\frac{\partial \boldsymbol{\epsilon}_{\theta}(\mathbf{x}_{t_i},\mathbf{z},t)}{\partial \mathbf{z}} + \mathcal{O}(\tilde{\mathbf{a}}_{\mathbf{x}}(t_i)-\mathbf{a}_{\mathbf{x}}(t_i) \\
&= \mathbf{a}_{\mathbf{z}}(t_{i+1}) + \mathcal{O}(h_{max}^2) + \mathcal{O}(\tilde{\mathbf{a}}_{\mathbf{x}}(t_i)-\mathbf{a}_{\mathbf{x}}(t_i)).
\end{aligned}
\tag{C.8}
$$

Repeat, this argument, from $\tilde{\mathbf{a}}_{\mathbf{z}}(t_0) = \mathbf{0}$ then we find

$$
\tilde{\mathbf{a}}_{\mathbf{z}}(t_M) = \mathbf{a}_{\mathbf{z}}(T) + \mathcal{O}(Mh_{max}^2) = \mathbf{a}_{\mathbf{z}}(T) + \mathcal{O}(h_{max}).
\tag{C.9}
$$

An identical argument can be constructed for $\mathbf{a}_{\theta}$, thereby finishing the proof. $\square$

## C.4 Proof of Theorem 2.1 when $k = 2$

We prove the discretization error of the AdjointDEIS-2M solver. Note that for the AdjointDEIS-2M solver we have $h_i = \lambda_{t_{i+1}} - \lambda_{t_{i-1}}$ and $\rho_i = \frac{\lambda_{t_i}-\lambda_{t_{i-1}}}{h_i}$. Furthermore, let $\Delta_i = \|\tilde{\mathbf{a}}_{\mathbf{x}}(t_i)-\mathbf{a}_{\mathbf{x}}(t_i)\|$. Without loss of generality, we will prove this only for $\mathbf{a}_{\mathbf{x}}$; the derivation for $\mathbf{a}_{\mathbf{z}}$ and $\mathbf{a}_{\theta}$ is analogous.

*Proof.* First, we consider the case of the adjoint state $\mathbf{a}_{\mathbf{x}}(t)$. Recall that the AdjointDEIS-2, see Equation (B.24), solver for $\mathbf{a}_{\mathbf{x}}$ with higher-order error terms is given by

$$
\mathbf{a}_{\mathbf{x}}(t_{i+1}) = \frac{\alpha_{t_i}}{\alpha_{t_{i+1}}}\mathbf{a}_{\mathbf{x}}(t_i) + \frac{1}{\alpha_{t_{i+1}}}\left[\frac{\sigma_{t_{i+1}}}{\alpha_{t_{i+1}}}(e^{h_i}-1)\mathbf{V}(\mathbf{x};t_i) + \frac{\sigma_{t_{i+1}}}{\alpha_{t_{i+1}}}(e^{h_i}-h_i-1)\mathbf{V}^{(1)}(\mathbf{x};t_i)\right] + \mathcal{O}(h_i^3).
\tag{C.10}
$$

Taylor's expansion yields

$$
\left\|\mathbf{a}_{\mathbf{x}}(t_{i+1}) - \left(\frac{\alpha_{t_i}}{\alpha_{t_{i+1}}}\mathbf{a}_{\mathbf{x}}(t_i) + \frac{1}{\alpha_{t_{i+1}}}\left[\frac{\sigma_{t_{i+1}}}{\alpha_{t_{i+1}}}(e^{h_i}-1)\mathbf{V}(\mathbf{x};t_i) + \frac{\sigma_{t_{i+1}}}{\alpha_{t_{i+1}}}(e^{h_i}-h_i-1)\mathbf{V}^{(1)}(\mathbf{x};t_i)\right]\right)\right\| \le Ch_i^3,
\tag{C.11}
$$

where $C$ is a constant that depends on $\mathbf{V}^{(2)}(\mathbf{x}_t,t)$. Also note that

$$
\left\|\mathbf{V}^{(1)}(\mathbf{x};t_i) - \frac{1}{\rho_i h_i}\big(\mathbf{V}(\mathbf{x};t_i) - \mathbf{V}(\mathbf{x};t_{i-1})\big)\right\| \le Ch_i.
\tag{C.12}
$$

Since $\rho_i$ is bounded away from zero, and $e^{-h_i} = 1 - h_i + h_i^2/2 + \mathcal{O}(h_i^3)$, we know

$$
\begin{aligned}
&\left\|(e^{h_i}-h_i+1)\mathbf{V}^{(1)}(\mathbf{x};t_i) - \frac{e^{h_i}-1}{2\rho_i}\big(\tilde{\mathbf{V}}(\mathbf{x};t_i) - \tilde{\mathbf{V}}(\mathbf{x};t_{i-1})\big)\right\| \\
&\le CLh_i(\Delta_i + \Delta_{i-1}) + Ch_i^3 + \frac{1}{\rho_i}\left|\frac{e^{h_i}-1}{2} - \frac{e^{h_i}-h_i-1}{h_i}\right|\|\mathbf{V}(\mathbf{x};t_i)-\mathbf{V}(\mathbf{x};t_{i-1})\| \\
&\le CLh_i(\Delta_i + \Delta_{i-1}) + Ch_i^3 + Ch_i^2\|\mathbf{V}(\mathbf{x};t_i)-\mathbf{V}(\mathbf{x};t_{i-1})\| \\
&\le CLh_i(\Delta_i + \Delta_{i-1}) + CM_i h_i^3,
\end{aligned}
\tag{C.13}
$$

where $M_i = 1 + \sup_{t_i \le t \le t_{i+1}}\|\mathbf{V}^{(1)}(\mathbf{x};t)\|$ and $L$ are the Lipschitz constants of $\mathbf{V}(\mathbf{x};t)$ by Theorem C.1. Then, $\Delta_{i+1}$ can be estimated as

$$
\begin{aligned}
\Delta_{i+1} &\le \frac{\alpha_{t_i}}{\alpha_{t_{i+1}}}\Delta_i + \frac{\sigma_{t_{i+1}}}{\alpha_{t_{i+1}}^2}L\Delta_i + \frac{\sigma_{t_{i+1}}}{\alpha_{t_{i+1}}^2}(CM_i h_i^3 + CLh_i(\Delta_i+\Delta_{i+1})) + Ch_i^3 \\
&\le \frac{\alpha_{t_i}}{\alpha_{t_{i+1}}}\Delta_i + \tilde{C}h_i(\Delta_i + \Delta_{i+1} + h_i^2).
\end{aligned}
\tag{C.14}
$$

Thus, $\Delta_{i+1} = \mathcal{O}(h_{max}^2)$ as long as $h_{max}$ is sufficiently small and $\Delta_0 + \Delta_1 = \mathcal{O}(h_{max}^2)$, which can be verified via Taylor expansion, thereby finishing the proof. $\square$

# D Proof of Theorem 2.2

For additional clarity, we let $\mathbf{x}(t) \equiv \mathbf{x}_t$ and likewise, $\mathbf{z}(t) \equiv \mathbf{z}_t$.

*Proof.* Recall that $\mathbf{z}(t)$ is a piecewise function of time with partitions of the time domain given by $\Pi = \{0 = t_0 < t_1 < \cdots < t_n = T\}$. Without loss of generality we consider some time interval $[t_{m-1}, t_m]$ for some $1 \leq m \leq n$. Consider the augmented state defined on the interval $\pi$:

$$\frac{\mathrm{d}}{\mathrm{d}t} \begin{bmatrix} \mathbf{x} \\ \mathbf{z} \end{bmatrix}(t) = \boldsymbol{f}_{\mathrm{aug}} = \begin{bmatrix} \boldsymbol{f}_\theta(\mathbf{x}(t), \mathbf{z}(t), t) \\ \overrightarrow{\partial}\,\mathbf{z}(t) \end{bmatrix}, \tag{D.1}$$

where $\overrightarrow{\partial}\,\mathbf{z} : [0, T] \to \mathbb{R}^z$ denotes the right derivative of $\mathbf{z}$ at time $t$. Let $\mathbf{a}_{\mathrm{aug}}$ denote the associated augmented adjoint state

$$\mathbf{a}_{\mathrm{aug}}(t) := \begin{bmatrix} \mathbf{a}_\mathbf{x} \\ \mathbf{a}_\mathbf{z} \end{bmatrix}(t). \tag{D.2}$$

The Jacobian of $\boldsymbol{f}_{\mathrm{aug}}$ has the form

$$\frac{\partial \boldsymbol{f}_{\mathrm{aug}}}{\partial[\mathbf{x}, \mathbf{z}]} = \begin{bmatrix} \frac{\partial \boldsymbol{f}_\theta(\mathbf{x}, \mathbf{z}, t)}{\partial \mathbf{x}} & \frac{\partial \boldsymbol{f}_\theta(\mathbf{x}, \mathbf{z}, t)}{\partial \mathbf{z}} \\ \mathbf{0} & \mathbf{0} \end{bmatrix}. \tag{D.3}$$

As the conditional information $\mathbf{z}(t)$ evolves with $\overrightarrow{\partial}\,\mathbf{z}(t)$ on $[t_{m-1}, t_m]$ in the forward flow of time. The derivative of the $\overrightarrow{\partial}\,\mathbf{z}$ w.r.t. $\mathbf{z}$ is clearly $\mathbf{0}$ as $\overrightarrow{\partial}\,\mathbf{z}$ is a function only of time $t$. Remark, that as the bottom row of the Jacobian $\boldsymbol{f}_{\mathrm{aug}}$ is all $\mathbf{0}$ and $\boldsymbol{f}_\theta$ is continuous in $t$ we can consider the evolution of $\mathbf{a}_{\mathrm{aug}}$ over the whole interval $[0, T]$ rather than just the partition $[t_{m-1}, t_m]$. Using Equations (D.2) and (D.3) we can define the evolution of the adjoint augmented state on $[0, T]$ as

$$\frac{\mathrm{d}\mathbf{a}_{\mathrm{aug}}}{\mathrm{d}t}(t) = -[\mathbf{a}_\mathbf{x} \quad \mathbf{a}_\mathbf{z}](t) \frac{\partial \boldsymbol{f}_{\mathrm{aug}}}{\partial[\mathbf{x}, \mathbf{z}]}(t). \tag{D.4}$$

Therefore, $\mathbf{a}_\mathbf{z}(t)$ evolves with the ODE

$$\mathbf{a}_\mathbf{z}(T) = 0, \qquad \frac{\mathrm{d}\mathbf{a}_\mathbf{z}}{\mathrm{d}t}(t) = -\mathbf{a}_\mathbf{x}(t)^\top \frac{\partial \boldsymbol{f}_\theta(\mathbf{x}(t), \mathbf{z}(t), t)}{\partial \mathbf{z}(t)}. \tag{D.5}$$

We have thus shown the evolution of $\mathbf{a}_\mathbf{z}(t)$ for some continuously differentiable function $\mathbf{z}(t)$.

Now we prove the solution is unique and exists. As $\mathbf{x}(t)$ is continuous and $\boldsymbol{f}_\theta$ is continuously differentiable in $\mathbf{x}$, it follows that the map $t \mapsto \frac{\partial \boldsymbol{f}_\theta}{\partial \mathbf{x}}(\mathbf{x}(t), \mathbf{z}(t), t)$ is a continuous function on the compact set $[0, T]$, and therefore it is bounded by some $L > 0$. Correspondingly, for $\mathbf{a}_\mathbf{x} \in \mathbb{R}^d$ it follows that the map $(\mathbf{a}_\mathbf{x}, t) \mapsto -\mathbf{a}_\mathbf{x}^\top \frac{\partial \boldsymbol{f}_\theta}{\partial[\mathbf{x}, \mathbf{z}]}(\mathbf{x}(t), \mathbf{z}(t), t)$ is Lipschitz in $\mathbf{a}_\mathbf{x}$ with Lipschitz constant $L$ and this constant is independent of $t$. Therefore, by the Picard-Lindelölf theorem the solution $\mathbf{a}_\mathbf{z}(t)$ exists and is unique. $\qquad\square$

## D.1 Continuous-time Extension of Discrete Conditional Information

The proof above makes some fairly lax assumptions about $\mathbf{z}(t)$. While we assumed the process was càdlàg and had right derivatives this restraints are too loose for adaptive step-size solvers. While this is not an issue for our AdjointDEIS, Kidger et al. [58] pointed out that an adaptive solver would take a long time to compute the backward pass as it would have to slow down at the discontinuities. One could follow the approach taken in [58] and construct natural cubic splines from a fully observed, but irregularly sampled time series $\{\mathbf{z}_{t_i}\}_{i=1}^N$ with $0 = t_0 < \cdots < t_n = T$. Then define $\mathbf{Z} : [0, T] \to \mathbb{R}^z$ as the natural cubic spline with knots at $t_0, \ldots, t_n$ such that $\mathbf{Z}(t_i) = \mathbf{z}_{t_i}$.

## D.2 Connection to Neural CDEs

Kidger et al. [58, Theorem C.1] showed that any equation of the form

$$\mathbf{x}_t = \mathbf{x}_0 + \int_0^t \boldsymbol{h}_\theta(\mathbf{x}_s, \mathbf{z}_s, s) \, \mathrm{d}s, \tag{D.6}$$

can be rewritten as a neural *controlled differential equation* (CDE) of the form

$$\mathbf{x}_t = \mathbf{x}_0 + \int_0^t \boldsymbol{f}_\theta(\mathbf{x}_s, s) \, \mathrm{d}\mathbf{z}_s, \tag{D.7}$$

where $\int \mathrm{d}\mathbf{z}_s$ is the Riemann-Stieltjes integral. *N.B.*, the converse is not true.

While there a certainly benefits to modeling with a neural CDE over a neural ODE, diffusion models in particular pre-train the model in the neural ODE sense. Moreover, we are also interested in updating $\mathbf{z}_{t_i}$ at each $t_i$ via gradient descent to solve our optimization problem.

# E   Details on Adjoints for SDEs

In this section, we provide further details on the continuous adjoint equations for diffusion SDEs that we omitted from the main paper due to their technical nature and for the purpose of brevity.

Consider the Itô integral given by

$$\mathbf{x}_T = \int_0^T \mathbf{x}_t \, \mathrm{d}\mathbf{w}_t, \tag{E.1}$$

where $\mathbf{x}_t$ is a continuous semi-martingale adapted to the filtration generated by the Wiener process $\{\mathbf{w}_t\}_{t \in [0,T]}$, $\{\mathcal{F}_t\}_{t \in [0,T]}$. The following quantity, however, is not defined

$$\int_T^0 \mathbf{x}_t \, \mathrm{d}\mathbf{w}_t. \tag{E.2}$$

This is because $\mathbf{x}_t$ and $\mathbf{w}_t$ are adapted to $\{\mathcal{F}_t\}_{t \in [0,T]}$ which is defined in forwards time. This means $\mathbf{x}_t$ does not anticipate future events only depends on *past* events. While this is generally sufficient when we wish to integrate backwards in time, we want *future* events to inform *past* events.

## E.1   Stratonovich Symmetric Integrals and Two-sided Filtration

Clearly, we need a different tool to model this backwards SDE. As such, taking inspiration from the work on neural SDEs [59], we follow the treatment of Kunita [60] for the forward and backward Fisk-Stratonovich integrals using *two-sided filtration*. Let $\{\mathcal{F}_{s,t}\}_{s \le t; s,t \in [0,T]}$ be a two-sided filtration, where $\mathcal{F}_{s,t}$ is the $\sigma$-algebra generated by $\{\mathbf{w}_v - \mathbf{w}_u : s \le u \le v \le t\}$ for $s,t \in [0,T]$ such that $s \le t$.

**Forward time.** For a continuous semi-martingale $\{\mathbf{x}_t\}_{t \in [0,T]}$ adapted to the forward filtration $\{\mathcal{F}_{0,t}\}_{t \in [0,t]}$, the Stratonovich stochastic integral is given as

$$\int_0^T \mathbf{x}_t \circ \mathrm{d}\mathbf{w}_t = \lim_{|\Pi| \to 0} \sum_{k=1}^N \frac{\mathbf{x}_{t_k} + \mathbf{x}_{t_{k-1}}}{2} (\mathbf{w}_{t_k} - \mathbf{w}_{t_{k-1}}) \tag{E.3}$$

where $\Pi = \{0 = t_0 < \cdots < t_N = T\}$ is a partition of the interval $[0,T]$ and $|\Pi| = \max_k t_k - t_{k-1}$. The forward filtration $\{\mathcal{F}_{0,t}\}_{t \in [0,t]}$ is analogous to the filtration defined in the prior section; therefore, any continuous semi-martingale adapted to it only considers *past* events and does not anticipate *future* events.

**Reverse time.** Consider the backwards Wiener process $\breve{\mathbf{w}}_t = \mathbf{w}_t - \mathbf{w}_T$ that is adapted to the backward filtration $\{\mathcal{F}_{s,T}\}_{s \in [0,T]}$, then for a continuous semi-martingale $\{\breve{\mathbf{x}}_t\}_{t \in [0,T]}$ adapted to the backward filtration, the backward Stratonovich integral is

$$\int_0^T \breve{\mathbf{x}}_t \circ \mathrm{d}\breve{\mathbf{w}}_t = \lim_{|\Pi| \to 0} \sum_{k=1}^N \frac{\breve{\mathbf{x}}_{t_k} + \breve{\mathbf{x}}_{t_{k-1}}}{2} (\breve{\mathbf{w}}_{t_{k-1}} - \breve{\mathbf{w}}_{t_k}) \tag{E.4}$$

The backward filtration $\{\mathcal{F}_{s,T}\}_{s \in [0,T]}$ is the opposite of the forward filtration in the sense that continuous semi-martingales adapted to it only depend on *future* events and do not anticipate *past* events. As such, time is effectively reversed.

**Remark E.1.** *While the Stratonovich symmetric integrals give us a powerful tool for integrating forwards and backwards in time with stochastic integrals, it is important that we use the **same** realization of the Wiener process.*

## E.2 Stochastic Flow of Diffeomorphisms

Consider the Stratonovich SDE defined as

$$\mathbf{x}_T = \mathbf{x}_0 + \int_0^T \boldsymbol{f}(\mathbf{x}_t, t)\, \mathrm{d}t + \int_0^T \boldsymbol{g}(\mathbf{x}_t, t) \circ \mathrm{d}\mathbf{w}_t, \tag{E.5}$$

where $\boldsymbol{f}, \boldsymbol{g} \in \mathcal{C}_b^{\infty,1}$, *i.e.*, they belong to the class of functions with infinitely many bounded derivatives w.r.t. the state and bounded first derivatives w.r.t. time. Thus, the SDE has a unique strong solution. Given a realization of the Wiener process, there exists a smooth mapping $\Phi$ called the *stochastic flow* such that $\Phi_{s,t}(\mathbf{x}_s)$ is the solution at time $t$ of the process described in Equation (E.5) started at $\mathbf{x}_s$ at time $s \le t$. This then defines a collection of continuous maps $\mathcal{S} = \{\Phi_{s,t}\}_{s \le t; s,t \in [0,T]}$ from $\mathbb{R}^d$ to itself.

Kunita [60, Theorem 3.7.1] shows that with probability 1 this collection $\mathcal{S}$ satisfies the flow property

$$\Phi_{s,t}(\mathbf{x}_s) = \Phi_{u,t}(\Phi_{s,u}(\mathbf{x}_s)) \quad s \le u \le t, \mathbf{x}_s \in \mathbb{R}^d, \tag{E.6}$$

and that each $\Phi_{s,t}$ is a smooth diffeomorphism from $\mathbb{R}^d$ to itself. Hence, $\mathcal{S}$ is the stochastic flow of diffeomorphisms generated by Equation (E.5). Moreover, the backward flow $\check{\Psi}_{s,t} := \Phi_{s,t}^{-1}$ satisfies the backwards SDE:

$$\check{\Psi}_{s,t}(\mathbf{x}_t) = \mathbf{x}_t - \int_s^t \boldsymbol{f}(\check{\Psi}_{u,t}(\mathbf{x}_t), u)\, \mathrm{d}u - \int_s^t \boldsymbol{g}(\check{\Psi}_{u,t}(\mathbf{x}_t), u) \circ \mathrm{d}\check{\mathbf{w}}_u, \tag{E.7}$$

for all $s, t \in [0, T]$ such that $s \le t$. This formulation makes intuitive sense as the *backwards* SDE differs only from the *forwards* SDE by a negative sign.

## E.3 Continuous Adjoint Equations

Now consider the adjoint flow $\mathbf{A}_{s,t}(\mathbf{x}_s) = \partial \mathcal{L}(\Phi_{s,t}(\mathbf{x}_s))/\partial \mathbf{x}_s$, then $\check{\mathbf{A}}_{s,t}(\mathbf{x}_t) = \mathbf{A}_{s,t}(\check{\Psi}_{s,t}(\mathbf{x}_t))$. Li et al. [59] show that $\check{\mathbf{A}}_{s,t}(\mathbf{x}_t)$ satisfies the backward SDE:

$$\check{\mathbf{A}}_{s,t}(\mathbf{x}_t) = \frac{\partial \mathcal{L}}{\partial \mathbf{x}_t} + \int_s^t \check{\mathbf{A}}_{u,t}(\mathbf{x}_t) \frac{\partial \boldsymbol{f}}{\partial \mathbf{x}_u}(\check{\Psi}_{u,t}(\mathbf{x}_t), u)\, \mathrm{d}u + \int_s^t \check{\mathbf{A}}_{u,t}(\mathbf{x}_t) \frac{\partial \boldsymbol{g}}{\partial \mathbf{x}_u}(\check{\Psi}_{u,t}(\mathbf{x}_t), u) \circ \mathrm{d}\check{\mathbf{w}}_u. \tag{E.8}$$

As the drift and diffusion coefficient of this SDE are in $\mathcal{C}_b^{\infty,1}$, the system has a unique strong solution.

## E.4 Proof of Theorem 3.1

We are now ready to put all of this together to prove the result from the main paper.

*Proof.*

$$\mathrm{d}\mathbf{x}_t = \boldsymbol{f}(\mathbf{x}_t, t)\, \mathrm{d}t + \boldsymbol{g}(t) \circ \mathrm{d}\mathbf{w}_t. \tag{E.9}$$

By Equation (E.8) the adjoint state admitted by the flow of diffeomorphisms generated by Equation (E.9) evolves with the SDE

$$\check{\mathbf{A}}_{s,t}(\mathbf{x}_t) = \frac{\partial \mathcal{L}}{\partial \mathbf{x}_t} + \int_s^t \check{\mathbf{A}}_{u,t}(\mathbf{x}_t) \frac{\partial \boldsymbol{f}}{\partial \mathbf{x}_u}(\check{\Psi}_{u,t}(\mathbf{x}_t), u)\, \mathrm{d}u + \underbrace{\int_s^t \check{\mathbf{A}}_{u,t}(\mathbf{x}_t) \frac{\partial \boldsymbol{g}}{\partial \mathbf{x}_u}(u) \circ \mathrm{d}\check{\mathbf{w}}_u}_{=\mathbf{0}}$$

$$= \frac{\partial \mathcal{L}}{\partial \mathbf{x}_t} + \int_s^t \check{\mathbf{A}}_{u,t}(\mathbf{x}_t) \frac{\partial \boldsymbol{f}}{\partial \mathbf{x}_u}(\check{\Psi}_{u,t}(\mathbf{x}_t), u)\, \mathrm{d}u. \tag{E.10}$$

Clearly, the adjoint state evolves with an ODE revolving around only the drift coefficient, *i.e.*, $\boldsymbol{f}$. Therefore, we can rewrite the evolution of the adjoint state as

$$\mathrm{d}\mathbf{a}_{\mathbf{x}}(t) = -\mathbf{a}_{\mathbf{x}}(t)^\top \frac{\partial \boldsymbol{f}}{\partial \mathbf{x}_t}(\mathbf{x}_t, t)\, \mathrm{d}t. \tag{E.11}$$

$\square$

### E.5 Converting the Itô SDE to Stratonovich

The diffusion SDE in Equation (3.1) is defined as an Itô SDE. However, Theorem 3.1 is defined as Stratonovich SDEs. However, an Itô SDE can be easily converted into the Stratonovich form, *i.e.*, for some Itô SDE of the form

$$d\mathbf{x}_t = \boldsymbol{f}(\mathbf{x}_t, t)\, dt + \boldsymbol{g}(\mathbf{x}_t, t)\, d\mathbf{w}_t \tag{E.12}$$

with a differentiable function $\sigma$, there exists a corresponding Stratonovich SDE of the form

$$d\mathbf{x}_t = \left[\boldsymbol{f}(\mathbf{x}_t, t) + \frac{1}{2}\frac{\partial \boldsymbol{g}}{\partial \mathbf{x}}(\mathbf{x}_t, t) \cdot \boldsymbol{g}(\mathbf{x}_t, t)\right] dt + \boldsymbol{g}(\mathbf{x}_t, t) \circ d\mathbf{w}_t. \tag{E.13}$$

As Equation (3.1) is defined such that $\boldsymbol{g}(\mathbf{x}_t, t) = g(t)$ and is independent of the state $\mathbf{x}_t$, then the SDE may be written in Stratonovich form as

$$d\mathbf{x}_t = \left[f(t)\mathbf{x}_t + \frac{g^2(t)}{\sigma_t}\boldsymbol{\epsilon}_\theta(\mathbf{x}_t, \mathbf{z}, t)\right] dt + g(t) \circ d\bar{\mathbf{w}}_t. \tag{E.14}$$

## F  Additional Experiments

In this section, we include some additional experiments which did not fit within the main paper.

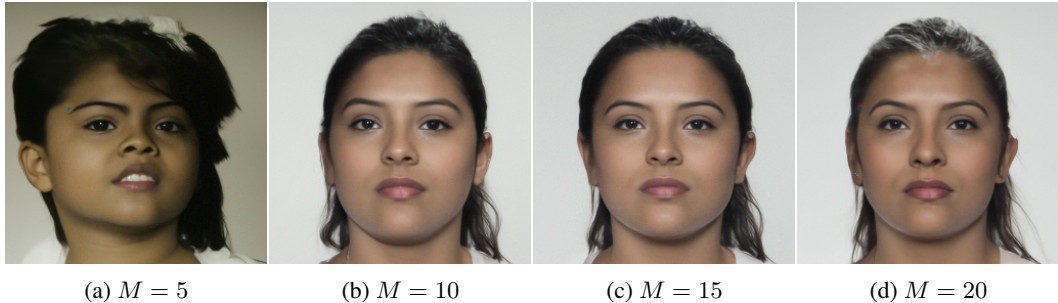

| (a) $M = 5$ | (b) $M = 10$ | (c) $M = 15$ | (d) $M = 20$ |

Figure 4: Morphed faces created by guided generation with AdjointDEIS with differing number of discretization steps.

### F.1  Impact of Discretization Steps

One of the advantages of AdjointDEIS is that the solver for the diffusion ODE and continuous adjoint equations are distinct. This means that we do not have to force $N = M$ enabling greater flexibility when using AdjointDEIS. As such, we explore the impact of using fewer steps to estimate the gradient while keeping the number of sampling steps $N = 20$ fixed. In Figure 4 we illustrate the impact of the change in the number of discretization steps when estimating the gradients. Unsurprisingly, the fewer steps we take, the less accurate the gradients are. This matches the empirical data presented in Table 4 which measures the impact of the performance of face morphing measured in MMPMR.

Table 4: Impact of number of discretization steps, $M$, on face morphing with AdjointDEIS. FMR = 0.1%, $\eta = 0.1$.

|  | MMPMR(↑) | | |
| --- | --- | --- | --- |
| $M$ (↓) | **AdaFace** | **ArcFace** | **ElasticFace** |
| 15 | **94.89** | 90.59 | **94.07** |
| 10 | 94.27 | 91.21 | 92.84 |
| 05 | 69.94 | 60.74 | 64.21 |

To explore the degradation in performance further. As Kidger [25] points out, due to recalculating the solution trajectory of the underlying state $\mathbf{x}_t$ backwards can differ, in not significant ways from the $\mathbf{x}_t$ calculated in the forward pass due to truncation errors. This is especially true for non-algebraically reversible solvers [61], although these can suffer from poor regions of stability. Consider the toy

example of the ODE $\dot{x}(t) = \lambda x(t)$, where $\lambda < 0$ and some numerical ODE solver. During the forward solve, most numerical ODE solvers with a non-trivial region of stability will find a reasonably useful solution as the errors decay exponentially; however, during the backward solve any small error is magnified exponentially instead.

One possible solution to this is to use interpolated adjoints wherein the solution states, $\mathbf{x}_t$, but not the internal states of $\boldsymbol{f}_\theta$ are stored. The backward solve can then interpolate between them as needed. Kim et al. [56] report that interpolated adjoints performed well on a stiff differential equation. In the case when $N = M$ it is quite convenient to simply store $\{\mathbf{x}_{t_i}\}_{i=1}^m$ during the forward solve. Moreover, in our use case where $N$ is quite small this adds little overhead. In Table 5 we compare recording the solution states vs finding them via solving DDIM forwards in time. We observe that AdjointDEIS performed better when using the recorded states.

Table 5: Recording the solution states vs backwards solve with DDIM. $M = 20, \eta = 0.001$. FMR = 0.1%.

| Record | MMPMR($\uparrow$) | | |
| --- | --- | --- | --- |
| | **AdaFace** | **ArcFace** | **ElasticFace** |
| ✗ | 94.27 | 89.78 | 93.87 |
| ✓ | 95.5 | 92.64 | 95.91 |

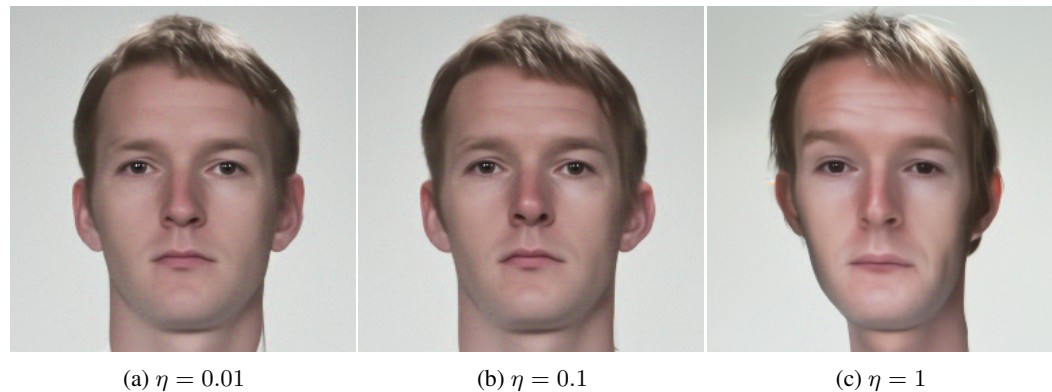

(a) $\eta = 0.01$        (b) $\eta = 0.1$        (c) $\eta = 1$

Figure 5: Morphed faces created by guided generation with AdjointDEIS with different learning rates. All used $M = 20$ the ODE variant.

### F.2    Impact of Learning Rate

We measure the impact of the learning rate on guided generation with AdjointDEIS in Table 4. Unsurprisingly, high learning rates lower performance, especially for less accurate gradients. *I.e.*, when $M$ is small. We illustrate an example of the impact in Figure 5. Clearly, the learning rate of $\eta = 1$ starts to distort the images even if it still fools the FR system.

### F.3    Number of Steps

As alluded to in the main paper, one of the drawbacks of diffusion SDEs is that they require small step sizes to work properly. We observe that the missing high frequency content is added back in when the step size is increased, see Figure 6.

## G    Implementation Details

### G.1    AdjointDEIS-2M Algorithm

For completeness we have the full AdjointDEIS-2M solver implemented in Algorithm 1 for solving the continuous adjoint equations for diffusion ODEs. We assume that there is another solver that

Table 6: Impact of learning rate, $\eta$, on face morphing with AdjointDEIS. FMR = 0.1%.

| SDE | $\eta$ | $M (\downarrow)$ | MMPMR($\uparrow$) | | |
|---|---|---|---|---|---|
| | | | **AdaFace** | **ArcFace** | **ElasticFace** |
| ✗ | 1 | 20 | 98.77 | 98.98 | 98.77 |
| ✗ | 0.1 | 20 | **99.8** | 98.77 | **99.39** |
| ✗ | 0.01 | 20 | 95.5 | 92.64 | 95.91 |
| ✗ | 1 | 10 | 50.92 | 49.69 | 50.92 |
| ✗ | 0.1 | 10 | 94.27 | 91.21 | 92.84 |
| ✗ | 1 | 05 | 2.66 | 2.04 | 1.84 |
| ✗ | 0.1 | 05 | 69.94 | 60.74 | 64.21 |
| ✓ | 1 | 20 | 98.57 | **99.59** | 98.98 |
| ✓ | 0.1 | 20 | 98.57 | 97.96 | 97.75 |

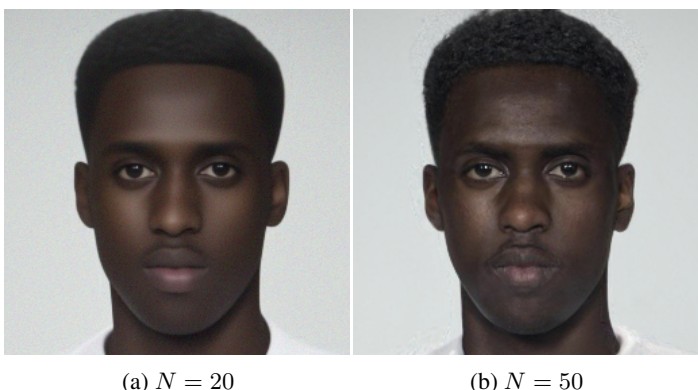

(a) $N = 20$        (b) $N = 50$

Figure 6: Morphed faces created by guided generation with AdjointDEIS with different number of sampling steps. SDE solver, $M = N$.

solves the backward ODE to yield $\{\tilde{\mathbf{x}}_{t_i}\}_{i=0}^M$. Remark that $\mathbf{a}_{\mathrm{aug}} \coloneqq [\mathbf{a}_\mathbf{x}, \mathbf{a}_\mathbf{z}, \mathbf{a}_\theta]$. Also Algorithm 1 can be used to solve the continuous adjoint equations for diffusion SDEs by simply adding the factor of 2 into the update equations.

## G.2 Code

Our code for AdjointDEIS will be available here at https://github.com/zblasingame/AdjointDEIS.

## G.3 Repositories Used

For reproducibility purposes, we provide a list of links to the official repositories of other works used in this paper.

1. The SYN-MAD 2022 dataset used in this paper can be found at https://github.com/marcohuber/SYN-MAD-2022.

2. The ArcFace models, MS1M-RetinaFace dataset, and MS1M-ArcFace dataset can be found at https://github.com/deepinsight/insightface.

3. The ElasticFace model can be found at https://github.com/fdbtrs/ElasticFace.

4. The AdaFace model can be found at https://github.com/mk-minchul/AdaFace.

5. The official Diffusion Autoencoders repository can be found at https://github.com/phizaz/diffae.

6. The official MIPGAN repository can be found at https://github.com/ZHYYYYYYYYYYYY/MIPGAN-face-morphing-algorithm.

**Algorithm 1** AdjointDEIS-2M.

---

**Require:** Initial values $\mathbf{a_x}(0)$, monotonically increasing time steps $\{t_i\}_{i=0}^M$, and noise prediction model $\boldsymbol{\epsilon}_\theta(\mathbf{x}_t, \mathbf{z}, t)$.

1: Denote $h_i := \lambda_{t_{i+1}} - \lambda_{t_i}$, for $i = 0, \ldots, M-1$.

2: $\tilde{\mathbf{a}}_\mathbf{x}(t_0) \leftarrow \mathbf{a_x}(0)$           $\triangleright$ Initialize an empty buffer $Q$.

3: $\tilde{\mathbf{a}}_\mathbf{z}(t_0) \leftarrow \mathbf{0}, \tilde{\mathbf{a}}_\theta(t_0) \leftarrow \mathbf{0}$.

4: $Q \overset{\text{buffer}}{\longleftarrow} [\tilde{\mathbf{V}}(\mathbf{x}, t_0), \tilde{\mathbf{V}}(\mathbf{z}, t_0), \tilde{\mathbf{V}}(\theta, t_0)]$

5: $\tilde{\mathbf{a}}_\mathbf{x}(t_1) \leftarrow \frac{\alpha_{t_i}}{\alpha_{t_1}} \tilde{\mathbf{a}}_\mathbf{x}(t_0) + \sigma_{t_1}(e^{h_0} - 1)\frac{\alpha_{t_0}^2}{\alpha_{t_1}^2} \tilde{\mathbf{a}}_\mathbf{x}(t_0)^\top \frac{\partial \boldsymbol{\epsilon}_\theta(\tilde{\mathbf{x}}_{t_0}, \mathbf{z}, t_0)}{\partial \tilde{\mathbf{x}}_{t_0}}$

6: $\tilde{\mathbf{a}}_\mathbf{z}(t_1) \leftarrow \tilde{\mathbf{a}}_\mathbf{z}(t_0) + \sigma_{t_1}(e^{h_0} - 1)\frac{\alpha_{t_0}}{\alpha_{t_1}} \tilde{\mathbf{a}}_\mathbf{x}(t_0)^\top \frac{\partial \boldsymbol{\epsilon}_\theta(\tilde{\mathbf{x}}_{t_0}, \mathbf{z}, t_0)}{\partial \mathbf{z}}$

7: $\tilde{\mathbf{a}}_\theta(t_1) \leftarrow \tilde{\mathbf{a}}_\theta(t_0) + \sigma_{t_1}(e^{h_0} - 1)\frac{\alpha_{t_0}}{\alpha_{t_1}} \tilde{\mathbf{a}}_\mathbf{x}(t_0)^\top \frac{\partial \boldsymbol{\epsilon}_\theta(\tilde{\mathbf{x}}_{t_0}, \mathbf{z}, t_0)}{\partial \theta}$

8: $Q \overset{\text{buffer}}{\longleftarrow} [\tilde{\mathbf{V}}(\mathbf{x}, t_1), \tilde{\mathbf{V}}(\mathbf{z}, t_1), \tilde{\mathbf{V}}(\theta, t_1)]$

9: **for** $i \leftarrow 1, 2, \ldots, M-1$ **do**

10:    $\rho_i \leftarrow \frac{h_{i-1}}{h_i}$

11:    $\mathbf{D}_i \leftarrow \left(1 + \frac{1}{2\rho_i}\right)\tilde{\mathbf{V}}(\mathbf{x}; t_i) - \frac{1}{2\rho_i}\tilde{\mathbf{V}}(\mathbf{x}; t_{i-1})$

12:    $\mathbf{E}_i \leftarrow \left(1 + \frac{1}{2\rho_i}\right)\tilde{\mathbf{V}}(\mathbf{z}; t_i) - \frac{1}{2\rho_i}\tilde{\mathbf{V}}(\mathbf{z}; t_{i-1})$

13:    $\mathbf{F}_i \leftarrow \left(1 + \frac{1}{2\rho_i}\right)\tilde{\mathbf{V}}(\theta; t_i) - \frac{1}{2\rho_i}\tilde{\mathbf{V}}(\theta; t_{i-1})$

14:    $\tilde{\mathbf{a}}_\mathbf{x}(t_{i+1}) \leftarrow \frac{\alpha_{t_i}}{\alpha_{t_{i+1}}} \tilde{\mathbf{a}}_\mathbf{x}(t_i) + \frac{\sigma_{t_{i+1}}}{\alpha_{t_{i+1}}^2}(e^{h_i} - 1)\mathbf{D}_i$

15:    $\tilde{\mathbf{a}}_\mathbf{z}(t_{i+1}) \leftarrow \tilde{\mathbf{a}}_\mathbf{z}(t_i) + \frac{\sigma_{t_{i+1}}}{\alpha_{t_{i+1}}}(e^{h_i} - 1)\mathbf{E}_i$

16:    $\tilde{\mathbf{a}}_\theta(t_{i+1}) \leftarrow \tilde{\mathbf{a}}_\theta(t_i) + \frac{\sigma_{t_{i+1}}}{\alpha_{t_{i+1}}}(e^{h_i} - 1)\mathbf{F}_i$

17:    **if** $i < M-1$ **then**

18:      $Q \overset{\text{buffer}}{\longleftarrow} [\tilde{\mathbf{V}}(\mathbf{x}, t_{i+1}), \tilde{\mathbf{V}}(\mathbf{z}, t_{i+1}), \tilde{\mathbf{V}}(\theta, t_{i+1})]$

19:    **end if**

20: **end for**

21: **return** $\tilde{\mathbf{a}}_\mathbf{x}(t_M), \tilde{\mathbf{a}}_\mathbf{z}(t_M), \tilde{\mathbf{a}}_\theta(t_M)$.

---

# H   Experimental Details

In this section, we outline the details for the experiments run in Section 5.

## H.1   DiM Algorithm

For completeness, we provide the DiM algorithm from [41] following the notation used in [35]. The original bona fide images are denoted $\mathbf{x}_0^{(a)}$ and $\mathbf{x}_0^{(b)}$. The conditional encoder is $\mathcal{E} : \mathcal{X} \to \mathcal{Z}$, $\Phi$ is the numerical diffusion ODE solver, $\Phi^+$ is the numerical diffusion ODE solver as time runs forwards from 0 to $T$. The algorithm is presented in Algorithm 2.

## H.2   NFE

In our reporting of the NFE we record the number of times the diffusion noise prediction U-Net is evaluated both during the encoding phase, $N_E$, and solving of the PF-ODE or diffusion SDE, $N$. We chose to report $N + N_E$ over $N + 2N_E$ as even though two bona fide images are encoded resulting in $2N_E$ NFE during encoding, this process can simply be batched together, reducing the NFE down to $N_E$. When reporting the NFE for the Morph-PIPE model, we report $N_E + BN$ where $B$ is the number of blends. While a similar argument can be made that the morphed candidates could be generated in a large batch of size $B$, reducing the NFE of the sampling process down to $N$, we chose to report $BN$ as the number of blends, $B = 21$, used in the Morph-PIPE is quite large, potentially resulting in Out Of Memory (OOM) errors, especially if trying to process a mini-batch of morphs. Using $N_E + N$ reporting over $N_E + BN$, the NFE of Morph-PIPE is 350, which is comparable to DiM. The reporting of NFE for AdjointDEIS was calculated as $N_E + n_{opt}(N + M)$ where $n_opt$

**Algorithm 2** DiM Framework.

---

**Require:** Blend parameter $w = 0.5$. Time schedule $\{t_i\}_{i=1}^N \subseteq [0, T], t_i < t_{i+1}$.

1: $\mathbf{z}_a \leftarrow \mathcal{E}(\mathbf{x}_0^{(a)})$          $\triangleright$ Encoding bona fides into conditionals.

2: $\mathbf{z}_b \leftarrow \mathcal{E}(\mathbf{x}_0^{(b)})$

3: **for** $i \leftarrow 1, 2, \ldots, N - 1$ **do**

4:      $\mathbf{x}_{t_{i+1}}^{(a)} \leftarrow \Phi^+(\mathbf{x}_{t_i}^{(a)}, \boldsymbol{\epsilon}_\theta(\mathbf{x}_{t_i}^{(a)}, \mathbf{z}_a, t_i), t_i)$ $\triangleright$ Solving the probability flow ODE as time runs from $0$ to $T$.

5:      $\mathbf{x}_{t_{i+1}}^{(b)} \leftarrow \Phi^+(\mathbf{x}_{t_i}^{(b)}, \boldsymbol{\epsilon}_\theta(\mathbf{x}_{t_i}^{(b)}, \mathbf{z}_b, t_i), t_i)$

6: **end for**

7: $\mathbf{x}_T^{(ab)} \leftarrow \text{slerp}(\mathbf{x}_T^{(a)}, \mathbf{x}_T^{(b)}; w)$          $\triangleright$ Morph initial noise.

8: $\mathbf{z}_{ab} \leftarrow \text{lerp}(\mathbf{z}_a, \mathbf{z}_b; w)$          $\triangleright$ Morph conditionals.

9: **for** $i \leftarrow N, N - 1, \ldots, 2$ **do**

10:      $\mathbf{x}_{t_{i-1}}^{(ab)} \leftarrow \Phi(\mathbf{x}_{t_i}^{(ab)}, \boldsymbol{\epsilon}_\theta(\mathbf{x}_{t_i}^{(ab)}, \mathbf{z}_{ab}, t_i), t_i)$     $\triangleright$ Solving the probability flow ODE as time runs from $T$ to $0$.

11: **end for**

12: **return** $\mathbf{x}_0^{(ab)}$

---

is the number of optimization steps and $M$ is the number of discretization steps for the continuous adjoint equations.

### H.3 Hardware

All of the main experiments were done on a single NVIDIA Tesla V100 32GB GPU. On average, the guided generation experiments for our approach took between 6 - 8 hours for the whole dataset of face morphs with a batch size of 8. Some additional follow-up work for the camera-ready version used an NVIDIA H100 Tensor Core 80GB GPU with a batch size of 16.

### H.4 Datasets

The SYN-MAD 2022 dataset is derived from the Face Research Lab London (FRLL) dataset [52]. FRLL is a dataset of high-quality captures of 102 different individuals with frontal images and neutral lighting. There are two images per subject, an image of a "neutral" expression and one of a "smiling" expression. The ElasticFace [53] FR system was used to select the top 250 most similar pairs, in terms of cosine similarity, of bona fide images for both genders, resulting in a total of 489 bona fide image pairs for face morphing [51], as some pairs did not generate good morphs on the reference set; we follow this minimal subset.

### H.5 FR Systems

All three FR systems use the Improved ResNet (IResNet-100) architecture [62] as the neural net backbone for the FR system. The ArcFace model is a widely used FR system [41, 43, 45, 49]. It employs an additive angular margin loss to enforce intra-class compactness and inter-class distance, which can enhance the discriminative ability of the feature embeddings [48]. ElasticFace builds upon the ArcFace model by using an elastic penalty margin over the fixed penalty margin used by ArcFace. This change results in an FR system with state-of-the-art performance [53]. Lastly, the AdaFace model employs an adaptive margin loss by weighting the loss relative to an approximation of the image quality [54]. The image quality is approximated via feature norms and is used to give less weight to misclassified images, reducing the impact of "low" quality images on training. This improvement allows the AdaFace model to achieve state-of-the-art performance in FR tasks.

The AdaFace and ElasticFace models are trained on the MS1M-ArcFace dataset, whereas the ArcFace model is trained on the MS1M-RetinaFace dataset. *N.B.*, the ArcFace model used in the identity loss is not the same ArcFace model used during evaluation. The model used in the identity loss is an IResNet-100 trained on the Glint360k dataset [63] with the ArcFace loss. We use the cosine distance to measure the distance between embeddings from the FR models. All three FR systems require images of $112 \times 112$ pixels. We resize every image, post alignment from dlib which ensures

the images are square, to $112 \times 112$ using bilinear down-sampling. The image tensors are then normalized such that they take values in $[-1, 1]$. Lastly, the AdaFace FR system was trained on BGR images so the image tensor is shuffled from the RGB format to the BGR format.

# I Analytic Formulations of Drift and Diffusion Coefficients

For completeness, we show how to analytically compute the drift and diffusion coefficients for a linear noise schedule Ho et al. [1] in the VP scenario Song et al. [15]. With a linear noise schedule $\log \alpha_t$ is found to be

$$\log \alpha_t = -\frac{\beta_1 - \beta_0}{4} t^2 - \frac{\beta_0}{2} t \tag{I.1}$$

on $t \in [0, 1]$ with $\beta_0 = 0.1, \beta_1 = 20$, following Song et al. [15]. The drift coefficient becomes

$$f(t) = -\frac{\beta_1 - \beta_0}{2} t - \frac{\beta_0}{2} \tag{I.2}$$

and as $\sigma_t = \sqrt{1 - \alpha_t^2}$ we find

$$\frac{\mathrm{d}\sigma_t^2}{\mathrm{d}t} = \frac{\mathrm{d}}{\mathrm{d}t} \left[ 1 - \exp\left( -\frac{\beta_1 - \beta_0}{4} t^2 - \frac{\beta_0}{2} t \right)^2 \right]$$

$$= ((\beta_1 - \beta_0)t + \beta_0) \exp\left( -\frac{\beta_1 - \beta_0}{2} t^2 - 2\beta_0 t \right) \tag{I.3}$$

Therefore, the diffusion coefficient $g^2(t)$ is

$$g^2(t) = \underbrace{((\beta_1 - \beta_0)t + \beta_0) \exp\left( -\frac{\beta_1 - \beta_0}{2} t^2 - 2\beta_0 t \right)}_{\frac{\mathrm{d}\sigma_t^2}{\mathrm{d}t}}$$

$$+ \underbrace{((\beta_1 - \beta_0)t + \beta_0) \left[ 1 - \exp\left( -\frac{\beta_1 - \beta_0}{4} t^2 - \frac{\beta_0}{2} t \right)^2 \right]}_{-2\frac{\mathrm{d}\log \alpha_t}{\mathrm{d}t} \sigma_t^2} \tag{I.4}$$

Importantly, $\frac{\mathrm{d}\sigma_t}{\mathrm{d}t}$ does not exist at time $t = 0$, as $\sigma_t$ is discontinuous at that point, and so an approximation is needed when starting from this initial step. In practice, adding a small $\epsilon \ll 1$ to $t = 0$ should suffice.

