# OpenReview forum: "AdjointDEIS: Efficient Gradients for Diffusion Models"
_NeurIPS.cc/2024/Conference — NeurIPS 2024 poster_

### Official Review · Reviewer_1FE9 · 2024-07-10

**Soundness:** 4
**Presentation:** 3
**Contribution:** 3
**Rating:** 7
**Confidence:** 4

**Summary:**

This paper presents a way to optimize the initial seed or control parameters to a diffusion network to optimize a differentiable loss applied to the final samples drawn from a discretization of the diffusion SDE / probability flow ODE. Put differently the idea is to apply the white box integration trick from DPM / DPM++ paper, and the method from the neural ODE paper to differentiate through an ODE solver.

In addition, the paper presents methods for backpropagating through an SDE.

In terms of experiments the method is demonstrated on a face morphing application.

**Strengths:**

The paper is well written and the derivation is mostly clear. I did not go through section 4 for the derivation of the SDE Adjoint Diffusion SDE too closely since I am not very familiar with the Stratonovich stochastic integral.

The paper is also fairly comprehensive since it presents the derivations for both the probability flow formulation as well as the SDE formulation.

Overall I think the problem tackled in the paper is interesting and the solutions are elegant.

**Weaknesses:**

The experimental results seem to be weak.

This paper presents a way to control the final outcome from a diffusion sampler using a differentiable loss, but the only application demonstrated in the paper is on face-morphing where the NFE are not controlled across the baselines. Some of the baselines have fewer NFE and it's not possible to say conclusively whether they underperform because they are worse methods or just because they had less computational budget.

AdjointDEIS-2M is not even implemented and compared.

Also, a lot more compelling demonstrations could be shown of these new algorithms. For example, a number of "controlnet" type applications could have been tried to create an image that matches a sketch of an output.

**Questions:**

Mainly I would like to understand a little bit more about the rationale for selecting the face morph application selection, and whether the authors think the empirical comparisons in Table 1 are apples-to-apples given the different NFE for the methods.


Some more minor comments
- eq (3.1) has a factor of 2 missing.
- around eq(2.5) it might be good to explicitly clarify that while VP typically uses alpha^2 + sigma^2 = 1 ibut you are not making that assumption here.

**Limitations:**

Yes limitations have been addressed.

---

> ### Author Rebuttal · Authors · 2024-08-02
>
> We thank reviewer 1FE9 for the helpful questions and interest in our work. We are happy the reviewer found the paper clear and comprehensive tackling an interesting problem. We address the questions raised by the reviewer below. We hope our responses help address the questions and are happy to answer any additional questions.
>
> > Mainly I would like to understand a little bit more about the rationale for selecting the face morph application selection, and whether the authors think the empirical comparisons in Table 1 are apples-to-apples given the different NFE for the methods.
>
> This work actually originated from research we were conducting on the face morphing problem. A popular technique in that area is to use identity guidance for GAN-based models (Zhang et al., "MIPGAN—Generating Strong and High Quality Morphing Attacks Using Identity Prior Driven GAN," in IEEE Transactions on Biometrics, Behavior, and Identity Science, 2021) wherein the optimal latent vector $\mathbf{z}^*$ is found by minimizing the identity loss defined between the generated morphed face $g(\mathbf{z})$ and the original bona fide faces $\mathbf{x}^{(a)}, \mathbf{x}^{(b)}$ such that
> $$\mathbf{z}^* = \mathop{\rm arg min}_{\mathbf{z} \in \mathcal{Z}} \mathcal{L}(g(\mathbf{z}), \mathbf{x}^{(a)}, \mathbf{x}^{(b)})$$
> This is done using a gradient descent algorithm and thus requires $\frac{\partial \mathcal{L}(g(\mathbf{z}), \mathbf{x}^{(a)}, \mathbf{x}^{(b)}))}{\partial \mathbf{z}}$ which can be found through simple automatic differentiation tools.
> We were seeking to perform a similar guided generation procedure, but with diffusion models.
> It is during this process that we developed the AdjointDEIS algorithm.
> Beyond the AdjointDEIS algorithm developed out of the work on guided generation for the face morphing problem, we are also interested in experimental illustration of AdjointDEIS as we compare to the original **Di**ffusion **M**orphs (**DiM**) and a recent identity guided extension called Morph-PIPE.
>
> We believe the comparisons in Table 1 are fair despite the different NFE numbers. We first present a brief summary of the different DiM algorithms with details on the NFE calculation:
> * **DiM-A** The original DiM algorithm, the model uses 250 NFEs to encode the bona fide images $\mathbf{x}_0^{(a)}, \mathbf{x}_0^{(b)}$ into the initial noise $\mathbf{x}_T^{(a)}, \mathbf{x}_T^{(b)}$ and conditional information $\mathbf{z}\_{a}, \mathbf{z}\_{b}$. The model then takes 100 NFEs to generated the morphed image from the morphed noise and conditionals $\mathbf{x}\_T^{(ab)}$ and $\mathbf{z}\_{ab}$.
> * **Fast-DiM** An improvement on DiM-A that uses higher-order ODE solvers to reduce the NFE. Fast-DiM use 250 NFEs for encoding and 50 NFEs for sampling.
> * **Morph-PIPE** An improvement on DiM-A that generates 21 possible morphs using a blend of 21 different interpolations between $\mathbf{x}_T^{(a)}$ and $\mathbf{x}_T^{(b)}$, likewise with the conditional information. Due to the 21 candidate morphs, the NFE for sampling is $21 \cdot 100 = 2100$. The morph which minimizes the identity loss is then chosen.
> * **DiM + AdjointDEIS** We apply AdjointDEIS-1 to the original DiM algorithm. We used 250 NFEs for encoding. Due to the optimization procedure, we were able to reduce the sampling NFE during each iteration of the optimization procedure to 20. We used 50 optimization steps, resulting in a total sampling NFE of $20 \cdot 50 = 1000$.
> * **DiM + SDE-AdjointDEIS** We apply SDE-AdjointDEIS-1 solver to the original DiM algorithm. We used 250 NFEs for encoding. As prior work has noticed, due to the intricacies of discretizing the diffusion SDE, numerical SDE solvers often take more steps to achieve good performance than a numerical ODE solver. We used 50 sampling steps but 10 optimization steps for 500 total sampling NFEs.
>
> As discussed in the Fast-DiM paper, increasing the number of sampling/encoding steps, and thereby NFE, for DiM-A, Fast-DiM, or Morph-PIPE would not meaningfully increase the effectiveness of the morphing attack in terms of MMPMR. While the higher NFE of the AdjointDEIS methods does indicate an increased computation cost over that of DiM or Fast-DiM, simply increasing the NFE for DiM or Fast-DiM would not give them equivalent performance to AdjointDEIS. Our higher NFE is due to the computation of the adjoint diffusion ODE/SDE.
> We would like to emphasize that compared to Morph-PIPE, the other identity guided method, our approaches achieve *superior performance with fewer NFE*.
>
> > Some more minor comments
> >    * eq (3.1) has a factor of 2 missing.
> >   * around eq(2.5) it might be good to explicitly clarify that while VP typically uses alpha^2 + sigma^2 = 1 ibut you are not making that assumption here.
>
> If accepted, both of these changes will be incorporated into the camera-ready version.

---

### Official Review · Reviewer_oW3N · 2024-07-11

**Soundness:** 2
**Presentation:** 2
**Contribution:** 2
**Rating:** 5
**Confidence:** 3

**Summary:**

In this paper, the authors proposed an accelerated  method for differentials of pretrained diffusion models with respect to its latent valuables or parameters by making use of
1. the Taylor expansion of the log-SNR parameter, and
2. the exact integral formula of the derivatives related to the probability flow ODE, which they call Adjoint Diffusion ODE.

They applied this method to morphing of two generated images. As they summarized in Table 1, their results achieved good performance.

For more details:
- The setting
    - Data diffusion SDE $d\mathbf{x}_t = f(t) \mathbf{x}_t + g(t) d\mathbf{w}_t$
        - Its integral $q(\mathbf{x}_t|\mathbf{x}_0) = \mathcal{N}(\mathbf{x}_t|\alpha_t \mathbf{x}_0, \sigma_t^2)$ where $f(t), g(t)$ are related to $\alpha_t, \sigma_t$ by Eq.(2.5).
    - A pretrained diffusion model $\epsilon_\theta(\mathbf{x},\mathbf{z}, t)$
        - Its generating process: the probability flow ODE $\frac{d\mathbf{x}\_t}{dt} = f(t)\mathbf{x}\_t + \frac{g(t)^2}{2 \sigma_t}\epsilon_\theta(\mathbf{x}, \mathbf{z}, t)$
        - They name the velocity (the RHS of the ODE) as $\mathbf{h}_\theta(\mathbf{x}, \mathbf{z}, t)$
- The purpose
    - Calculation of gradients of $\mathcal{L}(\mathbf{x}_0(\mathbf{x}_T, \mathbf{z}, \theta))$ with respect to $\mathbf{z}$ or $\theta$
    - which are defined by ODEs in Eq.(3.3) and (3.4).
- The method
    - One can recover the derivatives based on a vector $\hat{\mathbf{a}}_t = \frac{\partial \mathcal{L}}{\partial \mathbf{y}_t}$ where $\mathbf{y}_t = \alpha_t^{-1}\mathbf{x}_t$.
    - For $t<s$, one can get $\hat{\mathbf{a}}_s = \hat{\mathbf{a}}_t + \int\_{\lambda_t}^{\lambda_s} ... d\lambda$ (Proposition 3.1).
    - Applying the Taylor expansion on ... with $\lambda$ around $\lambda_t$, and drop $O((\lambda_s - \lambda_t)^{k+1})$ term, we get a concrete relation between $\hat{\mathbf{a}}_s$ and $\hat{\mathbf{a}}_t$ (Eq.(3.11)),
    - which provides explicit (approximated) formulas for the derivatives.
- The empirical result
    - They apply the proposed method to morphing (Section 5 Experiments).

**Strengths:**

- **quality(+)** The explanations are well written. The authors provide concrete proofs on their theorems and experiments with real data.

**Weaknesses:**

- **significance(-)** It is difficult to understand in which respect the proposed method is superior.
- **clarity(-)** I am almost convinced by the explanations, however there is a concern as explained in Questions.

**Questions:**

- for significance improvements:
    - **Q1(significance)** In the theoretical sections, the authors apply omitting the higher order terms of the Taylor expansion, so the resultant method should be just an approximation. Why is it good performance in Table 1 compared to other methods nonetheless the method is just an approximation?
- for clarity improvements:
    - **Q1(clarity)** In the Adjoint Empirical Diffusion SDE cases, the integral results in Eq.(4.12) and (4.13) look very similar to ODE cases in Eq.(3.13) and (3.14). The only difference seem to be (doubled) 2nd term. Why such similarity happens?
    - **Q2(clarity)** In the beginning of section 4, the authors wrote their motivation to analyze SDE cases as it possibly improve the result, however, it turns to be that the SDE version works worse than the ODE cases. Why does it happens?

**Limitations:**

They leave comments on
1. Lack of conditional generation in numerical experiments
2. No application of parameter differentiation in numerical experiments

in the final page.

---

> ### Author Rebuttal · Authors · 2024-08-02
>
> We thank reviewer oW3N for the detailed questions and interest in our work. We are glad that the reviewer thought our work was well written. We address the questions raised by the reviewer below. We would be happy to provide additional clarification or to answer further questions about our work.
>
> > * for significance improvements:
> >      * **Q1(significance)** In the theoretical sections, the authors apply omitting the higher order terms of the Taylor expansion, so the resultant method should be just an approximation. Why is it good performance in Table 1 compared to other methods nonetheless the method is just an approximation?
>
> The reviewer is correct that the AdjointDEIS-1 and SDE-AdjointDEIS-1 solvers are numerical approximations of the true adjoint diffusion ODE/SDE. However, we believe these approximations are still quite useful even with large step sizes. The local truncation error for Eq. (3.10) is $\mathcal{O}(h^{k+1})$, for the first-order solver this error is $\mathcal{O}(h^2)$. Intuitively, with more discretization steps $N \to \infty$ then $h \to 0$ the error for the first-order solver will disappear. In practice we found that $N = 20$ steps for the adjoint ODE and $N = 50$ steps for the adjoint SDE seemed to work well. As other works have noted, numerical SDE solvers take more steps due to the difficulty in discretezing the stochastic integral.
>
> The improvement compared to previous works of DiM, Fast-DiM, and Morph-PIPE in Table 1 can be attributed to the identity guided nature of DiM + AdjointDEIS.
> While there are discretization errors, we posit that these errors are small enough that the estimated gradient is still useful for identity guidance.
> One of the strengths of our proposed work is that the adjoint diffusion ODE solver is decoupled from the sampling procedure for the diffusion ODE, that means we can choose different numerical precision levels for sampling and estimating the gradients.
> We note that while Morph-PIPE does identity guidance, it does so through a brute force search over a space of 21 possible morphs whereas we perform identity guidance through estimated gradients.
>
> > * for clarity improvements:
> >     * **Q1(clarity)** In the Adjoint Empirical Diffusion SDE cases, the integral results in Eq.(4.12) and (4.13) look very similar to ODE cases in Eq.(3.13) and (3.14). The only difference seem to be (doubled) 2nd term. Why such similarity happens?
>
> This similarity is actually one of our key insights! We observed that the adjoint diffusion SDE actually simplifies to an ODE.
> The factor of 2 is present because of the difference between the PF ODE
> $$    \frac{\mathrm d \mathbf{x}\_t}{\mathrm dt} = f(t)\mathbf{x}\_t + \frac{g^2(t)}{2\sigma\_t}\boldsymbol\epsilon\_\theta(\mathbf{x}\_t, \mathbf{z}, t) $$
> and the diffusion SDE
> $$    \mathrm d \mathbf{x}\_t= \bigg [f(t)\mathbf{x}\_t + \frac{g^2(t)}{\sigma\_t}\boldsymbol\epsilon\_\theta(\mathbf{x}\_t, \mathbf{z}, t) \bigg ]\mathrm dt + g(t) \circ \mathrm d \tilde{\mathbf{w}}\_t $$
> where that factor 2 differs between the ODE and drift term of the SDE.
> This means that we can use the *exact* same solvers for the adjoint ODE as the adjoint SDE with the *only* exception being the factor of 2!
> The only caveat being the underlying state $\mathbf{x}\_t$ still evolves with the backwards flow, Eq. (4.6), and uses a different solver.
>
> >  **Q2(clarity)** In the beginning of section 4, the authors wrote their motivation to analyze SDE cases as it possibly improve the result, however, it turns to be that the SDE version works worse than the ODE cases. Why does it happens?
>
> While recent work like Nie et al., "The Blessing of Randomness: SDE Beats ODE in General Diffusion-based Image Editing" (ICLR 2024) shows the promise of SDEs in image editing, the face morphing problem is unique.
> The goal is to fool an Face Recognition (FR) system into accepting one morphed face image as belonging to *two* identities.
> Recent work into the face morphing problem has shown that visual fidelity and morphing performance are not necessarily correlated, see the DiM and Fast-DiM paper.
> Upon visual inspection, the morphs produced using the adjoint diffusion SDE solver look more realistic and less smoothed to our eyes.
> In terms of morphing performance in terms of MMPMR, we find it to be about the same as the adjoint diffusion ODE morphs.
> As such we would argue that the SDE morphs are actually *slightly* better than ODE morphs as they have comparable MMPMR performance and better visual fidelity.
>
> If accepted, we plan on making these contributions more clear in the camera-ready version.

---

> > ### Comment · Reviewer_oW3N · 2024-08-12
> >
> > Thanks for the clear explanations.
> >
> > On **Q1(significance)**, I understand a little about the reason for good performance that is related to  the capability of direct gradient estimates based on proposed methods in this paper. I think it makes sense.
> >
> > On **Q1(clarity)**, I understood the reason for appearance of 2. This seems to be in the same reasons of 2 in denominator of the probability flow ODE.
> >
> > On **Q2(clarity)**, I think it would be making known contributions more clear as the author responded, and I believe it will be improved in the camera-ready version.
> >
> > Overall, I think my concerns are solved by the response, and I would like to raise my score a little.

---

### Official Review · Reviewer_jxa4 · 2024-07-12

**Soundness:** 3
**Presentation:** 4
**Contribution:** 3
**Rating:** 7
**Confidence:** 2

**Summary:**

The paper proposes an adjoint sensitivity method -- AdjointDEIS -- for efficiently calculating gradients of diffusion SDE models. Current methods for naive backpropagation rely on discrete adjoints which are memory intensive. The authors introduce an approach based on the stochastic adjoint sensitivity method to solve the gradients with respect to initial noise, conditional information, and model parameters. The authors develop custom solvers for the adjoint problems along with the proposed sensitivity method. The methods are validated on a face morphing problem.

**Strengths:**

* **Great Flow/Presentation**: The general presentation of the paper is well done! Especially starting off with a more straightforward setting of an ODE and extending them to SDEs
* **Specialized Solver**: Diffusion models are quite widespread and developing specialized solvers leveraging the structure of the problem demonstrates a practical application.

**Weaknesses:**

* **Limited Experimental Scope**: While the overall analysis and formulations are sound, the overall experimental validation for the method is limited to just a single experiment.

**Questions:**

1. Point 2 in contributions: Aren't the continuous adjoint sensitivity methods used in Neural SDEs already general-purpose enough to handle diffusion problems?

---

> ### Author Rebuttal · Authors · 2024-08-02
>
> We thank reviewer jxa4 for the interest in our work and the insightful comments. We are encouraged the reviewer found the extensions from diffusion ODEs to diffusion SDEs as a strength of the paper. Below we respond to the question raised by the reviewer. We hope this helps address the questions and are happy to respond to any further questions.
>
> > Point 2 in contributions: Aren't the continuous adjoint sensitivity methods used in Neural SDEs already general-purpose enough to handle diffusion problems?
>
> We agree with the reviewer that the continuous adjoint sensitivity methods used in Neural SDEs are general-purpose enough to be applied to diffusion SDEs; however, w'd like to emphasize that one of contributions as we listed in the paper is "To the best of our knowledge, AdjointDEIS is the first general back-propagation technique **for** diffusion models that use an SDE solver for sampling".
> That is, to the best of our knowledge we are the first group to explore using the method of adjoint sensitivity *explicitly for* diffusion SDEs.
> A significant insight in our work comes from exploiting the structure of diffusion ODEs/SDEs, where, in the VP case---this can also be shown in the Variance Exploding (VE) case---the adjoint diffusion SDE simplifies to an ODE!
> This insight would not straightforwardly be shown by simply applying the Neural SDEs methods on the diffusion SDEs without inspecting the diffusion term of the Stratonovich SDE.
>
> Another contribution of our work is using the specific structure of the adjoint diffusion ODEs/SDEs to transform the ODE/SDE via exponential integrators to a much simpler formulation, removing the error from the linear error term.

---

> > ### Comment · Reviewer_jxa4 · 2024-08-12
> >
> > Thank you for addressing this.
> >
> > While the overall paper is well written and theoretically sound, I agree with reviewer vPC5 that the experimental scope of this paper is quite limited (with just a single experiment). As such I will keep my recommendation of Accept (score 7)

---

### Official Review · Reviewer_vPC5 · 2024-07-13

**Soundness:** 3
**Presentation:** 2
**Contribution:** 2
**Rating:** 6
**Confidence:** 2

**Summary:**

AdjointDEIS uses the method of adjoint sensitivity to compute gradients of diffusion models, which is more efficient and less memory intensive, and robust to the injected noise. This work proposes efficient solvers for both the adjoint probability flow ODE and the adjoint diffusion SDE. Experiments demonstrate the efficacy of the solvers on guided generation for a face morphing problem.

**Strengths:**

- AdjointDEIS reduces the adjoint of the PF ODE to the problem of exponential integrators, which enables use of the vast amount of literature in the areas.
 - Obtaining accurate gradients wrt initial noise x_T and the latent z can open the door to possibly novel applications of diffusion models.

**Weaknesses:**

- The number of experimental applications of the method seem lacking (just one application to face morphing attacks is included). I’d encourage the authors to think about applications to data assimilation or inverse problems for instance.
- Since the primary contribution of the paper is solver, having a proof of convergence rates or empirical studies of convergence rate would be valuable additions to the paper.

**Questions:**

- It would be interesting to visually plot example forward and backward trajectories of the diffusion processes. What is the typical magnitude of the numerical errors that accumulate during the backward ODE solve?

---

> ### Author Rebuttal · Authors · 2024-08-06
>
> We thank reviewer vPC5 for the helpful comments and interest in our work. We agree with the reviewer that AdjointDEIS can be applied to more applications like inverse problems. Below we respond to the questions and concerns raised by the reviewer. We hope this addresses the questions and are happy to answer any further questions.
>
> > Since the primary contribution of the paper is solver, having a proof of convergence rates or empirical studies of convergence rate would be valuable additions to the paper.
>
> We fully agree with the reviewer. If accepted we will include a proof the AdjointDEIS-$k$ is a $k$-th order solver with some mild assumptions to ensure the vector Jacobian product is Lipschitz w.r.t. $\mathbf{a}_t$ and that the step size is not significantly large $h\_{max} = \max\_{1 \leq j \leq M} h_j = \mathcal{O}(1 / M)$. We provide an excerpt from the proof below. The proof roughly follows the structure of Appendix B.3 from Lu et al. "DPM-solver: A fast ODE solver for diffusion probabilistic model sampling in around 10 steps" (NeurIPS, 2022).
>
> *Proof.* The AdjointDEIS-1 solver with higher-order error terms is given by
> \begin{equation}
> \def\bfa{{\mathbf{a}}}
> \def\bfx{{\mathbf{x}}}
> \def\bfz{{\mathbf{z}}}
> \bfa_{t_{i+1}}=\frac{\alpha_{t_i}}{\alpha_{t_{i+1}}}\bfa_{t_i} + \alpha_{t_i}^2\sigma_{t_{i+1}}(e^h_i - 1)\bfa_{t_i}^\top \frac{\partial \boldsymbol\epsilon_\theta(\bfx_{t_i}, \bfz, t_i)}{\partial \bfx_{t_i}} + \mathcal{O}(h_i^2)
> \end{equation}
> where $k=1, t_i = t, t_{i+1} = s, h_i = \lambda_s - \lambda_t$. Let $\\{\tilde{\mathbf{a}}\_{t\_i}\\}\_{i=1}^M$ denote the sequence computed by AdjointDEIS-$k$.
> Since the vector Jacobian product is Lipschitz w.r.t. $\mathbf{a}\_t$, we can write
> \begin{equation}
> \tilde\bfa_{t_i}^\top \frac{\partial \boldsymbol\epsilon_\theta(\tilde\bfx_{t_i}, \bfz, t_i)}{\partial \tilde\bfx_{t_i}}=\bfa_{t_i}^\top \frac{\partial \boldsymbol\epsilon_\theta(\bfx_{t_i}, \bfz, t_i)}{\partial \bfx_{t_i}} + \mathcal{O}(\tilde\bfa_{t_i} - \bfa_{t_i})
> \end{equation}
> then
> \begin{align}
> \def\bfa{{\mathbf{a}}}
> \def\bfx{{\mathbf{x}}}
> \def\bfz{{\mathbf{z}}}
> \tilde\bfa_{t_{i+1}} &= \frac{\alpha_{t_i}}{\alpha_{t_{i+1}}}\tilde\bfa_{t_i} + \alpha_{t_i}^2\sigma_{t_{i+1}}(e^h_i - 1)\tilde\bfa_{t_i}^\top \frac{\partial \boldsymbol\epsilon_\theta(\tilde\bfx_{t_i}, \bfz, t_i)}{\partial \tilde\bfx_{t_i}}\\\\
>         &= \frac{\alpha_{t_i}}{\alpha_{t_{i+1}}}\tilde\bfa_{t_i} + \alpha_{t_i}^2\sigma_{t_{i+1}}(e^h_i - 1)\bigg(\bfa_{t_i}^\top \frac{\partial \boldsymbol\epsilon_\theta(\bfx_{t_i}, \bfz, t_i)}{\partial \bfx_{t_i}} + \mathcal{O}(\tilde\bfa_{t_{i}} - \bfa_{t_{i}})\bigg)\\\\
>         &= \frac{\alpha_{t_i}}{\alpha_{t_{i+1}}}\tilde\bfa_{t_i} + \alpha_{t_i}^2\sigma_{t_{i+1}}(e^h_i - 1)\bfa_{t_i}^\top \frac{\partial \boldsymbol\epsilon_\theta(\bfx_{t_i}, \bfz, t_i)}{\partial \bfx_{t_i}} + \mathcal{O}(\tilde\bfa_{t_{i}} - \bfa_{t_{i}})\\\\
>         &= \bfa_{t_i} + \mathcal{O}(h_{max}^2)  + \mathcal{O}(\tilde\bfa_{t_{i}} - \bfa_{t_{i}})
> \end{align}
> Repeating this argument, from $\tilde{\mathbf{a}}\_{t\_i} = \mathbf{a}\_0$ then
> \begin{equation}
> \def\bfa{{\mathbf{a}}}
> \def\bfx{{\mathbf{x}}}
> \def\bfz{{\mathbf{z}}}
> \tilde\bfa_{t_M} = \bfa_T + \mathcal{O}(Mh_{max}^2) = \bfa_T + \mathcal{O}(h_{max})
> \end{equation}
> Thus the global truncation error is $\mathcal{O}(h_{max})$ and thus completes the proof Q.E.D.
>
> > It would be interesting to visually plot example forward and backward trajectories of the diffusion processes. What is the typical magnitude of the numerical errors that accumulate during the backward ODE solve?
>
> The forward and backwards trajectories will look identical just with a reversal in time. The sampling trajectory of the probability flow ODE starts with white noise and slowly adds information back into a clean image. Likewise, the backwards ODE solver for the probability flow ODE takes the clean image and adds noise to it. The typical magnitude is numerically quite small as the gradients themselves have a small magnitude. From what I recall the magnitude of errors were about $10^{-5}$.

---

> > ### Comment · Reviewer_vPC5 · 2024-08-13
> > **Response to authors**
> >
> > Thank you for addressing my comment. I have updated my score. I still encourage the authors to do more extensive evaluation for the method in the future.

---

### Author Rebuttal · Authors · 2024-08-07

# General Response
We thank all the reviewers for all of their time and feedback on our submitted manuscript.

---

We are delighted to see that the reviewers appreciated the practical significance of our method, highlighting that it "can open the door to possibly novel applications of diffusion models." (**reviewer vPC5**) and "demonstrates a practical application" (**reviewer jxa4**).

Additionally, we are glad to see that the reviewers appreciated that we "start[ed] off with a more straightforward setting of an ODE and extending them to SDEs" (**reviewer jxa4**) and thought that "the problem tackled in the paper is interesting and the solutions are elegant" (**reviewer 1FE9**).

Lastly, we are pleased that the reviewers found the paper to be "well written" (**reviewers oW3N and 1FE9**) and "fairly comprehensive" (**reviewer 1FE9**).

---
We primarily address the concerns and questions raised by the reviewers in our individual responses; however, we address some common concerns below.

**Significance of Contribution.** In light of the feedback we received, we make the following improvements:
* Emphasizing the significance that the adjoint diffusion SDE simplifies to an ODE and is **identical** to the adjoint probability flow ODE with the exception of a factor of 2 on the vector Jacobian term.
* Prove that the AdjointDEIS-$k$ solvers are in fact $k$-th order solvers obtaining a global truncation error of $\mathcal{O}(h^k)$. For more details, please refer to the response to reviewer vPC5.
* Highlight that the calculation of the adjoint probability flow ODE is decoupled from the probability flow ODE, i.e., separate numerical solvers can be used with **different** step sizes! This means we could use **fewer** steps to obtain a working estimate of the gradient and still perform guided diffusion. The step size could be scheduled, whereas the loss decreases with each optimization iteration using the adjoint state the step size for the adjoint ODE solver can increase to reduce computation.

**Experimental Section.** We acknowledge that additional experimental results would help strength our manuscript. We believe that the example of face morphing provides a good illustration of the utility of AdjointDEIS in guided generation problems. Our hope is that the compelling motivation of AdjointDEIS coupled with the new theoretical results will provide a springboard for further research into guided generation with gradients from AdjointDEIS.

We hope that our responses to the reviewers convincingly address the reviewers' concerns and are happy to answer any further questions.

Sincerely,

The Authors

---

### Decision · Program_Chairs · 2024-09-25

**Decision:**

Accept (poster)

**Comment:**

This work leverages adjoint sensitivity analysis to efficiently compute gradients of the probability flow ODE implementation of diffusion model, in order to optimize latent variables and hyperparameters. Reviewers and AC agreed about the merits of this work, and acceptance is recommended.